# Accurate quantification of copy-number aberrations and whole-genome duplications in multi-sample tumor sequencing data

Simone Zaccaria [1] & Benjamin J. Raphael [1 ✉]

Copy-number aberrations (CNAs) and whole-genome duplications (WGDs) are frequent somatic mutations in cancer but their quantification from DNA sequencing of bulk tumor samples is challenging. Standard methods for CNA inference analyze tumor samples individually; however, DNA sequencing of multiple samples from a cancer patient has recently become more common. We introduce HATCHet (Holistic Allele-specific Tumor Copy-number Heterogeneity), an algorithm that infers allele- and clone-specific CNAs and WGDs jointly across multiple tumor samples from the same patient. We show that HATCHet outperforms current state-of-the-art methods on multi-sample DNA sequencing data that we simulate using MASCoTE (Multiple Allele-specific Simulation of Copy-number Tumor Evolution). Applying HATCHet to 84 tumor samples from 14 prostate and pancreas cancer patients, we identify subclonal CNAs and WGDs that are more plausible than previously published analyses and more consistent with somatic single-nucleotide variants (SNVs) and small indels in the same samples.

[1] Department of Computer Science, Princeton University, Princeton, NJ 08540, USA. ✉email: braphael@princeton.edu

Cancer results from the accumulation of somatic mutations in cells, yielding a heterogeneous tumor composed of distinct subpopulations of cells, or clones, with different complements of mutations[1]. Quantifying this intratumor heterogeneity and inferring past tumor evolution have been shown to be crucial in cancer treatment and prognosis[2–4]. CNAs are frequent somatic mutations in cancer that amplify or delete one or both the alleles of genomic segments, chromosome arms, or even entire chromosomes[5]. In addition, WGD, a doubling of all chromosomes, is a frequent event in cancer with an estimated frequency higher than 30% in recent pan-cancer studies[5–8]. Accurate inference of CNAs and WGDs is crucial for quantifying intratumor heterogeneity and reconstructing tumor evolution, even when analyzing only SNVs[9–13].

In principle, CNAs can be detected in DNA sequencing data by examining two signals: (1) the difference between the observed and expected counts of sequencing reads that align to a locus, quantified by the read-depth ratio (RDR), and (2) the proportion of reads belonging to the two distinct alleles of the locus, quantified by the B-allele frequency (BAF) of heterozygous germline single-nucleotide polymorphisms (SNPs). In practice, the inference of CNAs and WGDs from DNA sequencing data is challenging, particularly for bulk tumor samples that are mixtures of thousands-millions of cells. In such mixtures the signal from the observed reads is a superposition of the signals from normal cells and distinct tumor clones, which share the same clonal CNAs but are distinguished by different subclonal CNAs. One thus needs to deconvolve, or separate, this mixed signal into the individual components arising from each of these clones. This deconvolution is complicated as both the CNAs and the proportion of cells originating from each clone in the mixture are unknown; in general the deconvolution problem is underdetermined with multiple equivalent solutions. In the past few years, over a dozen methods have been developed to solve different versions of this copy-number deconvolution problem[6,9,14–27]. These methods rely on various simplifying assumptions, such as: only one tumor clone is present in the sample, no WGDs, etc. While these assumptions remove ambiguity in copy-number deconvolution, it is not clear that the resulting solutions are accurate, particularly in cases of highly aneuploid tumors.

Single-cell DNA sequencing[28] obviates the need for copy-number deconvolution, but remains a specialized technique with various technical and financial challenges, and thus is not yet widely used in sequencing of cancer patients, particularly in clinical settings. A valuable intermediate between DNA sequencing of single cells and DNA sequencing of a single bulk tumor sample is DNA sequencing of multiple bulk tumor samples from the same patient; these samples may be obtained from multiple regions of a primary tumor, matched primary and metastases, or longitudinal samples[11,12,26,29–31]. A number of approaches have demonstrated that simultaneous analysis of SNVs from multiple tumor samples helps to resolve uncertainties in clustering SNVs into clones[32,33] and to reduce ambiguities in inferring phylogenetic trees[11,29,34–36]. However, available methods for inferring CNAs analyze individual samples, losing the important information that multiple samples from the same patient share many CNAs that occurred during tumor evolution.

To slice through the thicket of ambiguity in copy-number deconvolution, we introduce HATCHet, an algorithm to infer allele- and clone-specific CNAs as well as the proportions of distinct tumor clones jointly across one or more samples from the same patient. HATCHet provides a fresh perspective on CNA inference and includes two main algorithmic innovations that address limitations of existing methods. First, HATCHet jointly analyzes multiple samples by globally clustering RDRs and BAFs along the entire genome and across all samples, and by solving a matrix factorization problem to infer allele- and clone-specific copy numbers from all samples. In contrast, existing methods[6,9,14–22,25–27] infer allele-specific copy numbers on each sample independently (with one exception[25]) and locally cluster RDRs and BAFs for neighboring regions in each sample separately. Second, HATCHet separates two distinct sources of ambiguity in the copy-number deconvolution problem, the presence of subclonal CNAs and the occurrence of WGDs, and uses a model-selection criterion to distinguish these sources. In contrast, existing methods attempt to fit a unique value for the variables tumor ploidy and purity (or equivalent variables) to the observed RDRs and BAFs, conflating different sources of ambiguity in the data.

In this paper, we evaluate HATCHet on both simulated and cancer data. We show that HATCHet outperforms six current state-of-the-art methods[9,15,17,21,22,25,27,37] on 256 samples from 64 patients simulated by MASCoTE, a framework that we develop to generate DNA sequencing data from multiple mixed samples with appropriate corrections for the differences in genome lengths between normal and tumor clones. Next, we apply HATCHet on whole-genome multi-sample DNA sequencing data from 49 samples from 10 metastatic prostate cancer patients[11] and 35 samples from four metastatic pancreas cancer patients[30]. We show that HATCHet's inferred subclonal CNAs and WGDs are more plausible than reported in published analyses and more consistent with somatic SNVs and small indels measured in the same samples, resulting in alternative reconstructions of tumor evolution and metastatic seeding patterns.

## Results

**HATCHet algorithm**. We introduce HATCHet, an algorithm to infer allele- and clone-specific copy numbers and clone proportions for several tumor clones jointly across multiple bulk tumor samples from the same patient (Fig. 1a). The inputs to HATCHet are RDRs and BAFs for short genomic bins across $k$ tumor samples from the same patient. We assume that each sample is a mixture of at most $n$ clones, including the normal diploid clone and one or more tumor clones with different CNAs. The goal is to infer these clones and their proportions in each sample, where we model the effects of CNAs as $m$ segments, or clusters of genomic bins. The two outputs of HATCHet are: (1) copy-number states $(a_{s,i}, b_{s,i})$ that indicate the allele-specific copy numbers $a_{s,i}$ and $b_{s,i}$ for each segment $s$ in each clone $i$, and that form two $m \times n$ matrices $A = [a_{s,i}]$ and $B = [b_{s,i}]$; and (2) clone proportions $u_{i,p}$ that indicate the fraction of cells in sample $p$ belonging to clone $i$, and that form an $n \times k$ matrix $U = [u_{i,p}]$.

HATCHet separates the inference of $A$, $B$, and $U$ into two modules. The first module infers the allele-specific fractional copy numbers $f_{s,p}^A = \sum_i a_{s,i} u_{i,p}$ and $f_{s,p}^B = \sum_i b_{s,i} u_{i,p}$ of each segment $s$ in each sample $p$, which form two $m \times k$ matrices $F^A$ and $F^B$. Specifically, this module has three steps. First, HATCHet computes RDRs and BAFs in short genomic bins (with a user adjustable size set to 50 kb in our analysis) along the genome (Fig. 1b). Second, HATCHet clusters RDRs and BAFs globally along the entire genome and jointly across all samples using a Bayesian non-parametric clustering algorithm[38] (Fig. 1c). This clustering leverages the fact that samples from the same patient have a shared evolutionary history. Finally, HATCHet aims to infer the fractional copy numbers $F^A$ and $F^B$. Importantly, $F^A$ and $F^B$ are not measured directly and must be inferred from the observed RDRs and BAFs. However, $F^A$ and $F^B$ are not identifiable from DNA sequencing data of bulk tumor samples and typically have multiple equally plausible values (Supplementary Fig. 1). We show in Methods that if one knows whether or not a WGD has occurred, then $F^A$ and $F^B$ are determined under

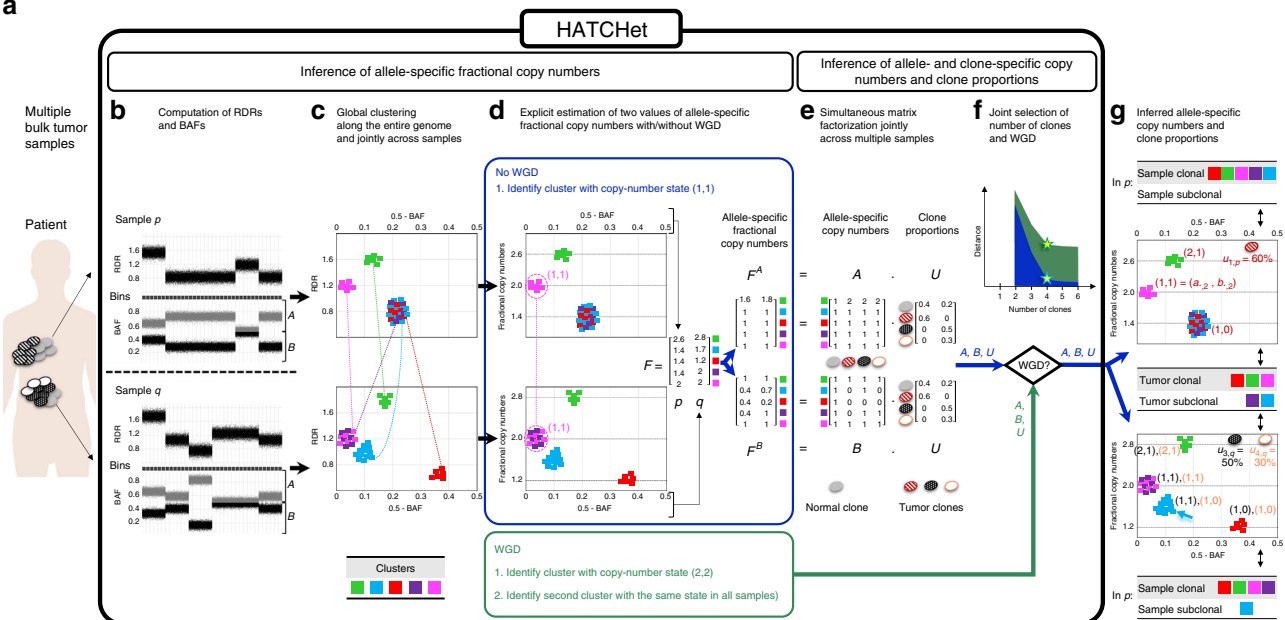

**Fig. 1 Overview of HATCHet algorithm. a** HATCHet takes in input DNA sequencing data from multiple bulk tumor samples of the same patient and has five steps. **b** First, HATCHet calculates the RDRs and BAFs in bins of the reference genome (black squares). Here, we show two tumor samples $p$ and $q$. **c** Second, HATCHet clusters the bins based on RDRs and BAFs globally along the entire genome and jointly across samples $p$ and $q$. Each cluster (color) includes bins with the same copy-number state within each clone present in $p$ or $q$. **d** Third, HATCHet estimates two values for the fractional copy number of each cluster by scaling RDRs. If there is no WGD, the identification of the cluster (magenta) with copy-number state (1, 1) is sufficient and RDRs are scaled correspondingly. If a WGD occurs, HATCHet identifies an additional cluster with identical copy-number state in all tumor clones. Dashed black horizontal lines in the scaled BAF-RDR plot represent values of fractional copy numbers that correspond to clonal CNAs. **e** Fourth, HATCHet factors the allele-specific fractional copy numbers $F^A$, $F^B$ into the allele-specific copy numbers $A$, $B$, respectively, and the clone proportions $U$. Here, there is a normal clone and 3 tumor clones. **f** Last, HATCHet's model-selection criterion identifies the matrices $A$, $B$, and $U$ in the factorization while evaluating the fit according to both the inferred number of clones and presence/absence of a WGD. **g** HATCHet outputs allele- and clone-specific copy numbers (with the color of the corresponding clone) and clone proportions (in the top right part of each plot) for each sample. Clusters are classified according to the inference of unique/different copy-number states in each sample (sample-clonal/subclonal) and across all tumor clones (tumor-clonal/subclonal).

few reasonable assumptions. Thus, in the third step HATCHet estimates two values of $F^A$ and $F^B$, assuming the presence or absence of a WGD, and defers the selection between these alternatives until after the inference of $A$, $B$, and $U$, i.e., the clonal composition (Fig. 1d).

The second module of HATCHet computes $A$, $B$, and $U$ from the inferred values of $F^A$ and $F^B$ by solving a matrix factorization problem. Since $F^A = AU$ and $F^B = BU$, the copy-number deconvolution problem corresponds to the problem of simultaneously factoring $F^A$ into the factors $A$, $U$ and $F^B$ into the factors $B$, $U$. In general, multiple factorizations may exist and thus HATCHet enforces additional constraints on the allowed factorizations, including a maximum copy number ($a_{s,i} + b_{s,i} \le c_{max}$), a minimum clone proportion (either $u_{i,p} \ge u_{min}$ or $u_{i,p} = 0$), and evolutionary relationships among the tumor clones. HATCHet solves the resulting optimization problem using a coordinate-descent algorithm (Fig. 1e). Finally, HATCHet uses a model-selection criterion to select the number $n$ of clones and the occurrence of WGD, as these values are unknown a priori and must be selected carefully to avoid overfitting the data. Specifically, HATCHet infers $A$, $B$, and $U$ for every value of $n$ and for the two values of $F^A$ and $F^B$ estimated in the first module. Then, HATCHet considers the trade-off between the inference of subclonal CNAs (resulting in higher $n$ and more clones present in a sample) and WGD to select the solution (Fig. 1f, g).

HATCHet differs from existing methods for copy-number deconvolution in a number of ways, which are summarized in Supplementary Table 1 and further detailed in Methods.

Importantly, HATCHet addresses the challenges of nonidentifiability and model selection using a different strategy than existing methods. Recognizing that the estimation of $F^A$, $F^B$ and their deconvolution into $A$, $B$, and $U$ are two different sources of ambiguity in the data, HATCHet defers the selection of $F^A$ and $F^B$ until after the deconvolution. This allows HATCHet to consider the trade-off between solutions with many subclonal CNAs vs. solutions with WGD (Supplementary Fig. 1). In contrast, existing methods[6,9,14–20,23–27] attempt to fit values for the variables tumor ploidy and purity (or equivalent variables) that best model the observed RDRs and BAFs. However, tumor ploidy and purity are composite variables that sum the contributions of the unknown copy numbers and proportions of multiple clones. Thus, tumor ploidy and purity are not ideal coordinates to evaluate tumor mixtures as many different clonal compositions may be equally plausible in these coordinates, particularly when more than one tumor clone is present or a WGD occurs (Supplementary Figs. 2 and 3).

**HATCHet outperforms existing methods for copy-number deconvolution.** We compared HATCHet with six current state-of-the-art methods for copy-number deconvolution, i.e., Battenberg[9], TITAN[17], THetA[21,22], cloneHD[25], Canopy[37] (with fractional copy numbers from FALCON[15]), and ReMixT[27], on simulated data. Most current studies that simulate DNA sequencing data from mixed samples containing CNAs do not account for the different genome lengths of distinct clones[15–17,25,39–44]; this oversight leads to incorrect simulation of read counts (Supplementary Figs. 4 and 5). Therefore, we introduce MASCoTE, a

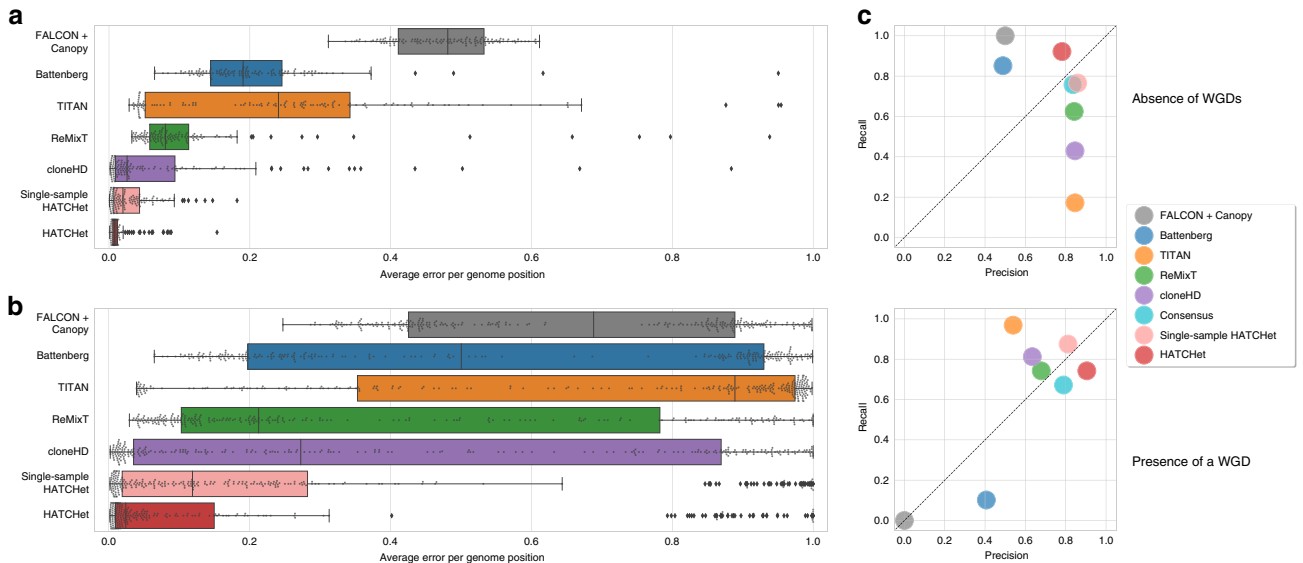

**Fig. 2 HATCHet outperforms existing methods in the inference of CNAs, their proportions, and WGDs. a** Average allele-specific error per genome position for the copy-number states and their proportions inferred by each method (here excluding THetA which does not infer allele-specific copy numbers) on 128 simulated tumor samples from 32 patients without a WGD, and where each method was provided with the true values of the main parameters (e.g., tumor ploidy, number of clones, and maximum copy number). HATCHet outperforms all the other methods even when it considers single samples individually (single-sample HATCHet). **b** Average allele-specific error per genome position on 256 simulated tumor samples from 64 patients, half with a WGD, and where each method infers all relevant parameters including tumor ploidy, number of clones, etc. HATCHet outperforms all the other methods, even when considering single samples individually (single-sample HATCHet). Box plots show the median and the interquartile range (IQR), and the whiskers denote the lowest and highest values within 1.5 times the IQR from the first and third quartiles, respectively. **c** Average precision and recall in the prediction of the absence of a WGD and the presence of a WGD in a sample. HATCHet is the only method with high precision and recall (>75%) in both the cases, even compared to a consensus of the other methods based on a prediction for majority. While Battenberg and Canopy underestimate the presence of WGDs (<20% and 0% recall), TITAN, ReMixT, and cloneHD overestimates the absence of WGDs (<20%, <62%, and <50% recall).

simulation framework to correctly generate DNA sequencing reads from multiple bulk tumor samples, with each sample containing one or more clones that share the same evolutionary history during which CNAs and/or WGD occur (Supplementary Fig. 6 and further details in Methods). We simulated DNA sequencing reads from 256 tumor samples (1–3 tumor clones) for 64 patients (3–5 samples per patient), half with a WGD and half without a WGD (Supplementary Fig. 7).

We separated the comparison of methods into two parts in order to assess both the inference of CNAs and proportions, as well as the prediction of WGDs. First, we provided the true values of the main parameters inferred by each method to assess the ability to retrieve the correct solution without the difficulty of model selection. Second, we ran each method in default mode. In both cases, we applied HATCHet jointly on all samples as well as separately on each sample (single-sample HATCHet) to quantify the contribution of the global clustering and the factorization model which capture the dependency across samples. Further details are in Supplementary Notes 1 and 2.

We first ran all methods on the 128 samples from 32 patients without a WGD and also providing the true value of the main parameters required for each method (e.g., tumor ploidy and number of clones). We found that HATCHet outperformed all other methods (Fig. 2a and Supplementary Figs. 8–12), demonstrating the advantages of HATCHet's joint analysis of multiple samples. While single-sample HATCHet has slightly worse performance, it also outperformed all other methods, suggesting that the additional features of HATCHet, such as the clustering of RDRs and BAFs along the entire genome, play an important role. Further discussions of these results are in Supplementary Note 3.

To assess the simultaneous prediction of WGD and inference of CNAs and proportions, we next ran the methods on all 256 samples from all 64 patients, requiring that each method infers all relevant parameters, including tumor ploidy and number of clones. Note that we excluded THetA from this comparison as it does not automatically infer presence/absence of WGDs. Not surprisingly, in this more challenging setting, all methods have lower performance, but HATCHet and single-sample HATCHet continue to outperform the other methods (Fig. 2b and Supplementary Figs. 13–17), even when assessing the prediction of amplified/deleted segments independently from the presence of a WGD (Supplementary Fig. 18). HATCHet is the only method with high (>75%) precision and recall in the inference of both presence and absence of WGD, while other methods tend to be biased towards presence or absence (Fig. 2c and Supplementary Fig. 19). We observed the same bias even when taking the consensus of the other methods, a procedure used in the recent PCAWG analysis of >2500 whole-cancer genomes[7] (Fig. 2c). The higher performance of HATCHet illustrates the advantage of performing model selection using the natural variables of the problem, i.e., the copy numbers $A$, $B$ and the clone proportions $U$, rather than selecting a unique solution based on tumor ploidy and purity as done by existing methods (Supplementary Fig. 20). Further discussions of these results are in Supplementary Note 4.

Finally, we further assessed the performance of HATCHet by comparing the copy-number profiles derived by HATCHet on whole-exome sequencing of bulk tumor samples with the copy-number profiles from DOP-PCR single-cell DNA sequencing from the same tumors. On 21 bulk tumor samples from 8 breast cancer patients[45,46], we observed a reasonable consistency between HATCHet's profiles and those from single cells (Supplementary Figs. 21 and 22). Additional details of this analysis are reported in Supplementary Note 5.

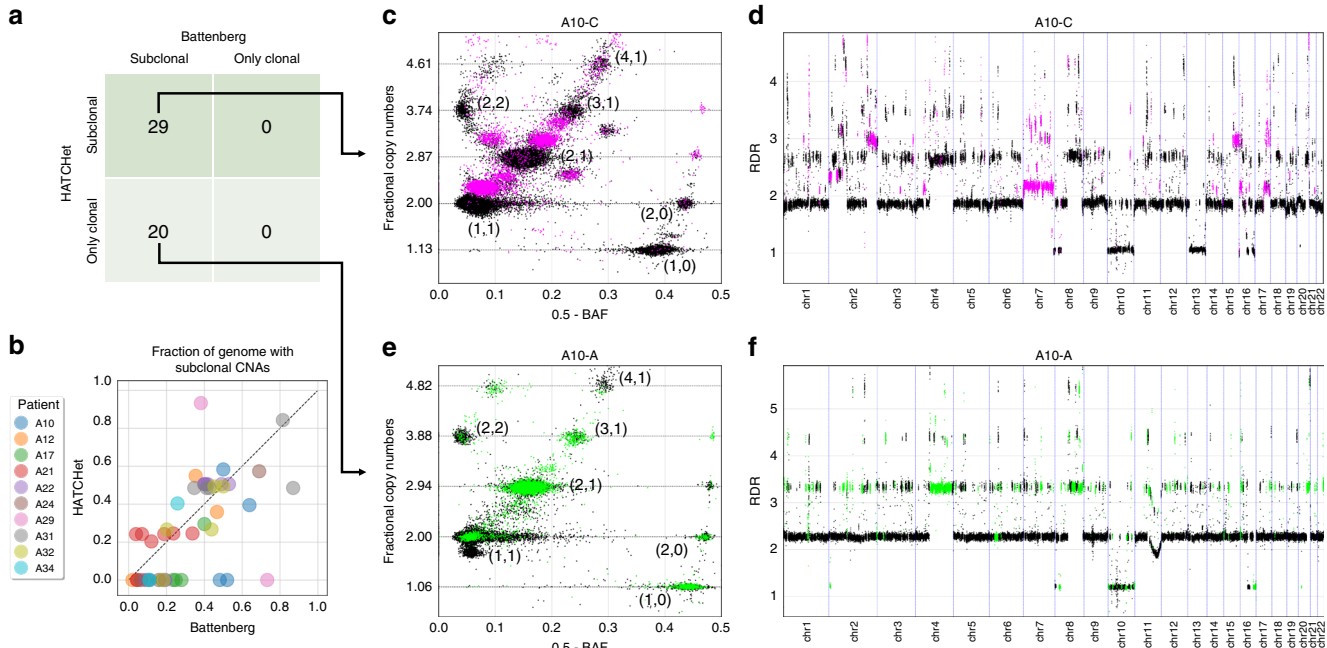

**Fig. 3 HATCHet identifies moderate amount of subclonal CNAs in prostate cancer patients. a** HATCHet identifies subclonal CNAs in 29 samples, while Battenberg identifies subclonal CNAs in all 49 samples. **b** In the 29 samples where both methods identify subclonal CNAs, HATCHet and Battenberg infer similar fractions of the genome with subclonal CNAs (dotted diagonal), while in the other 20 samples only Battenberg retrieves relatively high fractions of subclonal CNAs. **c** In sample A10-C of patient A10, both HATCHet and Battenberg identify reliable subclonal CNAs that correspond to sample-subclonal clusters (magenta) with clearly intermediate positions in the scaled BAF-RDR plot (each point corresponds to 50 kb genomic bin) between those of sample-clonal clusters (black clusters with corresponding copy-number states) with clonal CNAs (dashed black lines). **d** The sample-subclonal clusters in **c** correspond to large genomic regions (magenta) with values of RDR (for 50kb genomic bins) clearly distinct from the RDR values of regions from sample-clonal clusters (black). **e** In sample A10-A of patient A10, Battenberg identifies extensive clusters of 50kb genomic bins with subclonal CNAs (green). However, such clusters are not clearly distinguished in the scaled BAF-RDR plot from the sample-clonal clusters (black with corresponding copy-number states). HATCHet infers only clonal CNAs in this sample. **f** The sample-subclonal clusters in **e** correspond to large genomic regions (green) with values of RDR (for 50kb genomic bins) approximately equal to the RDR values of nearby regions from sample-clonal clusters (black).

**HATCHet identifies well-supported subclonal CNAs**. We used HATCHet to analyze two whole-genome DNA sequencing datasets with multiple tumor samples from individual patients (Supplementary Note 6): 49 primary and metastatic tumor samples from 10 prostate cancer patients[11] (Supplementary Fig. 23) and 35 primary and metastatic tumor samples from four pancreas cancer patients[30] (Supplementary Fig. 24). While both datasets contain multiple tumor samples from individual patients, the previously published analyses inferred CNAs in each sample independently. Moreover, these studies reached opposite conclusions regarding the landscape of CNAs in these tumors: Gundem et al.[11] reported subclonal CNAs in all prostate samples, while Makohon-Moore et al.[30] reported no subclonal CNAs in the pancreas samples. An important question is whether this difference is due to cancer-type specific or patient-specific differences in CNA evolution of these tumors, or a consequence of differences in the bioinformatic analyses. We investigated whether the HATCHet's analysis would confirm or refute the discordance between these studies.

On the prostate cancer dataset, HATCHet identified subclonal CNAs in 29/49 samples. In contrast, the published analysis[11] of these samples used Battenberg for CNA inference and identified subclonal CNAs in all 49 samples (Fig. 3a). On the 29 samples where both methods reported subclonal CNAs, we found that the two methods identified a similar fraction of the genome with subclonal CNAs (Fig. 3b). Moreover, on these samples, there are clear sample-subclonal clusters of genomic bins (i.e., with different copy-number states in the same sample, cf. Fig. 1g and Supplementary Fig. 25) with RDRs and BAFs that are clearly

distinct and intermediate between those of sample-clonal clusters (Fig. 3c). These sample-subclonal clusters correspond to subclonal CNAs affecting large genomic regions (Fig. 3d). In contrast, on the 20 samples where only Battenberg reported subclonal CNAs, the sample-subclonal clusters only identified by Battenberg do not have RDRs and BAFs that are clearly distinguishable from the sample-clonal clusters (Fig. 3e, f and Supplementary Figs. 26 and 27).

While it is possible that Battenberg has higher sensitivity in detecting subclonal CNAs than HATCHet, the extensive subclonal CNAs reported by Battenberg in all samples is concerning. This is because the inference of subclonal CNAs will always produce a better fit to the observed RDRs and BAFs, but with a cost of increasing the number of parameters required to describe the copy-number states (model complexity). Battenberg models the clonal composition of each segment independently (Supplementary Fig. 28), and thus has 6× more parameters than HATCHet on this dataset (Supplementary Fig. 29). To avoid overfitting, it is important to evaluate the trade-off between model fit and model complexity. Battenberg does not include a model-selection criterion to evaluate this trade-off, and it consequently infers a high fraction of subclonal CNAs in every sample (Supplementary Note 7) without fitting the observed RDRs and BAFs better than HATCHet (Supplementary Note 8). In contrast, HATCHet uses a model-selection criterion to identify the number of clones; consequently in 20/49 samples HATCHet infers that all the subclonal CNAs identified by Battenberg are instead clonal (Supplementary Fig. 30). Since HATCHet fits the observed RDRs and BAFs as well as Battenberg (Supplementary

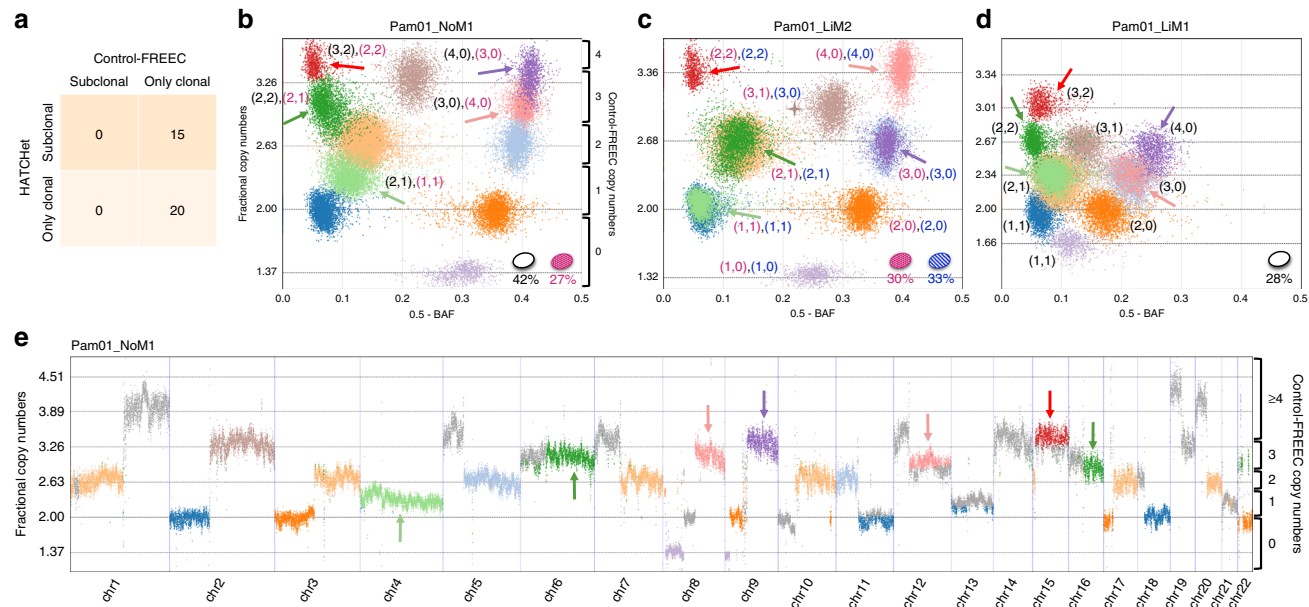

**Fig. 4 HATCHet identifies well-supported subclonal CNAs in metastatic pancreas cancer patients. a** HATCHet identifies subclonal CNAs in 15 of 35 samples, while published analysis used Control-FREEC and excluded subclonal CNAs. **b** In the lymph node metastasis sample Pam01_NoM1, HATCHet infers two distinct tumor clones (ellipses in lower right of plot with corresponding proportions) and a tumor purity of 69%. Five sample-subclonal clusters (arrows) of 50kb genomic bins occupy intermediate positions between the other sample-clonal clusters (dashed black lines) in the scaled BAF-RDR plot, and thus have distinct copy-number states in the two clones, corresponding to subclonal CNAs. Control-FREEC copy numbers are shown on the right y-axis labels. **c** In a second liver metastasis sample Pam01_LiM2 from the same patient, HATCHet infers two distinct tumor clones, one (red) shared with the lymph node sample Pam01_NoM1. A large sample-subclonal cluster (brown, starred) occupies an intermediate position in the scaled BAF-RDR plot and has distinct copy-number states in the two clones. In contrast, the five sample-subclonal clusters in Pam01_NoM1 (arrows) clearly overlap the sample-clonal clusters in this sample and thus correspond to clonal CNAs (dashed black lines). **d** In the liver metastasis sample Pam01_LiM1, HATCHet identifies a single tumor clone (white) that is shared with the lymph node metastasis sample Pam01_NoM1 in **b**. The five sample-subclonal clusters in Pam01_NoM1 (arrows) correspond to clonal CNAs in sample Pam01_LiM1 but have different copy-number states than those in **c**. The inferred low tumor purity (28%) of this sample results in a partial overlap of clusters that are clearly distinguished in higher purity samples in **b** and **c**. **e** The five sample-subclonal clusters in Pam01_NoM1 (arrows) correspond to large genomic regions with values of RDR that are clearly distinct from the other sample-clonal clusters (dashed black lines). Genomic regions that are part of small clusters or have out-of-scale values are reported in gray. Ranges of fractional copy numbers corresponding to the total copy numbers inferred by Control-FREEC in the previously published analysis are shown on the right y-axis labels.

Fig. 31) but without subclonal CNAs, the extensive subclonal CNAs reported by Battenberg in these samples are equally well-explained as clonal CNAs.

Finally, we found that ReMixT's inference of subclonal CNAs from the same dataset was more similar to HATCHet than Battenberg (Supplementary Fig. 32). Since both HATCHet and ReMixT outperformed Battenberg on the simulated data, the similarity between HATCHet and ReMixT on this dataset suggests that Battenberg's results are less accurate. Further details of this analysis are in Supplementary Note 9.

On the pancreas cancer dataset, HATCHet identified subclonal CNAs in 15/35 samples (Fig. 4a). In contrast, the published analysis[30] of these samples used Control-FREEC for CNA inference, which assumes that all CNAs are clonal and contained in all tumor cells in a sample (Supplementary Fig. 33). Overall, HATCHet reported a greater fraction of the genome with CNAs (Supplementary Fig. 34) and better fit the observed RDRs and BAFs (Supplementary Fig. 35) using less than 1/3 of the parameters used by Control-FREEC (Supplementary Fig. 36). The identification of subclonal CNAs is supported by the presence of sample-subclonal clusters which have RDRs and BAFs that are clearly distinct from those of sample-clonal clusters (Fig. 4b–d); moreover, many of these clusters correspond to large subclonal CNAs spanning chromosomal arms (Fig. 4e) and have different copy-number states across different samples (i.e., tumor-subclonal clusters, cf. Fig. 1g and Supplementary Fig. 25). HATCHet's joint analysis across multiple samples also aids in

the CNA inference in low-purity samples: the liver metastasis sample Pam01_LiM1 has an inferred tumor purity of 28% causing clusters of genomic bins with distinct copy-number states to have similar values of RDR and BAF (Fig. 4d). The distinct clusters are identified by leveraging the signal from a higher purity sample (Pam01_LiM2 in Fig. 4c).

HATCHet's joint analysis of multiple tumor samples from the same patient enables the direct identification of clones that are shared across multiple samples. Overall, HATCHet infers that 14/49 samples from the prostate cancer patients and 13/35 samples from the pancreas cancer patients have evidence of subclones shared between multiple samples, compared to 46/49 and 0/35 in the published copy-number analyses, respectively (Supplementary Note 10 and Supplementary Figs. 37–39). For example, HATCHet reports that the lymph node sample of pancreas cancer patient Pam01 (Fig. 4b) is a mixture of two clones with each of these clones present in exactly one of the two distinct liver metastases from the same patient (Fig. 4c, d). We found that the shared subclones identified by HATCHet are more consistent with previous SNV analyses[11,30]. Moreover, we found that the resulting metastatic seeding patterns agree with previous reports of limited heterogeneity across metastases[11,30] (Supplementary Note 11 and Supplementary Figs. 40–42) and provide evidence for polyclonal migrations in pancreas cancer patients (Supplementary Fig. 43), consistent with the reports of polyclonal migrations in mouse models of pancreatic tumors[47]. Finally, additional analyses of RDRs and BAFs further support the

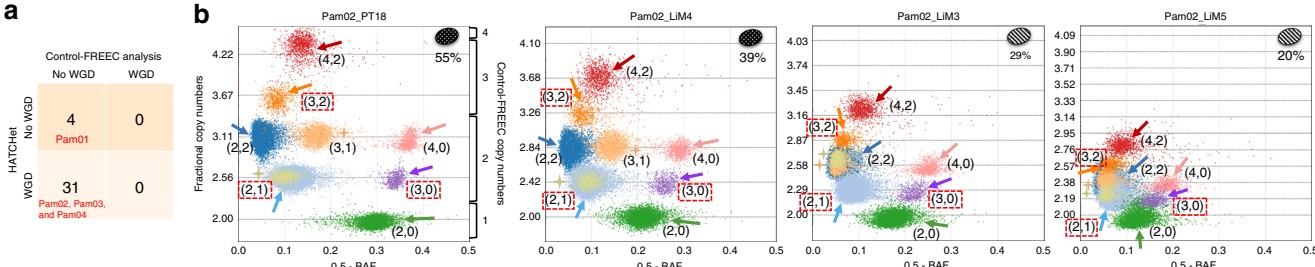

**Fig. 5 HATCHet identifies WGDs in three of four pancreas cancer patients. a** HATCHet predicts a WGD in all 31 samples from three patients (Pam02, Pam03, and Pam04). In contrast, published analysis used Control-FREEC and excluded WGDs. **b** In four samples of patient Pam02, HATCHet predicts a WGD and infers two tumor clones (ellipses in upper right of plot with corresponding proportions) with seven large tumor-clonal clusters (arrows with corresponding copy-number states). These clusters preserve their relative positions in the scaled BAF-RDR plot (each point corresponds to 50kb genomic bin) across samples and their fractional copy numbers correspond to sample-clonal clusters in each sample (dashed black lines), supporting the inference of a tumor-clonal CNA (i.e., unique copy-number state across samples) for each of these clusters. Note that without a WGD three clusters (red dashed squares) would correspond to subclonal CNAs in all samples. Two additional clusters (peach and olive, starred) are tumor-subclonal as they change their relative position across samples (Pam02_PT18 and Pam02_LiM4 vs. Pam02_LiM3 and Pam02_LiM5), supporting the inference of two distinct tumor clones in this patient. The total copy numbers inferred by Control-FREEC in published analysis are shown on the right y-axis labels in the first scaled BAF-RDR plot.

subclonal CNAs identified by HATCHet (Supplementary Note 12 and Supplementary Figs. 44 and 45).

**HATCHet reliably identifies WGDs.** We next examined the prediction of WGDs on the prostate and pancreas cancer datasets. The previously published analyses of these datasets reached opposite conclusions regarding the landscape of WGDs in these tumors: Gundem et al.[11] reported WGDs in 12 samples of 4 prostate cancer patients (A12, A29, A31, and A32), while Makohon-Moore et al.[30] did not evaluate the presence of WGDs in the pancreas cancer samples, despite reports of high prevalence of WGD in pancreas cancer[48]. We investigated whether HATCHet analysis would confirm or refute the different prevalence of WGDs reported in the previous studies.

On the prostate cancer dataset, there is strong agreement between WGD predictions from Battenberg and HATCHet, with discordance on only 2/49 samples (Supplementary Fig. 46a, b). Note that Battenberg does not explicitly state whether a WGD is present in a sample, and thus we used the criterion from previous pan-cancer analysis[5–8,12] that a tumor sample with ploidy >3 corresponds to WGD. Since Battenberg's solutions were manually chosen from different alternatives in the published analysis, the strong agreement between these predictions is a positive indicator for HATCHet's automated model selection. The two discordant samples, A12-C and A29-C, are single samples from patients A12 and A29, respectively. Battenberg predicted a WGD only in A12-C and no WGD in the other samples from A12. Conversely, Battenberg predicted no WGD in A29-C but a WGD in the other sample from A29. However, the divergent predictions of WGD are likely due to the Battenberg's independent analysis of each sample and is not well supported by the data. In contrast, HATCHet jointly analyzes multiple samples and predicts the absence/presence of a WGD consistently across all samples from the same patient (no WGD in A12 and a WGD in A29), providing simpler solutions with an equally good fit of the observed data (Supplementary Note 13 and Supplementary Figs. 46–47).

On the pancreas dataset, the published analysis excluded the possibility of WGDs and assumed that tumor ploidy is always equal to 2. Instead, HATCHet predicted a WGD in all 31 samples from three of the four patients (Fig. 5a). These results are consistent with recent reports of the high frequency of WGD (~45%) and massive rearrangements in pancreatic cancer[26,48], and also supported by additional analyses (Supplementary Fig.

48). All 31 samples from the three patients with a WGD display several large clusters of genomic regions with clearly distinct values of RDR and BAF. When jointly considering all samples from the same patient, these clusters are clearly better explained by the occurrence of a WGD (Fig. 5b) than by the presence of many subclonal CNAs, as the latter would result in the unlikely presence of distinct tumor clones with the same proportions in all samples (Supplementary Fig. 49). By directly evaluating the trade-off between subclonal CNAs and WGDs in the model selection, HATCHet makes more reasonable predictions of the occurrence of WGDs.

**HATCHet's CNAs better explain somatic SNVs and small indels.** We evaluated how well the copy numbers and proportions inferred by each method explain the observed read counts of somatic SNVs and small indels—two classes of mutations that were not used in the identification of CNAs. Specifically, we compared the observed variant-allele frequency (VAF) of each mutation with the best predicted VAF obtainable from the inferred copy-number states and proportions at the genomic locus (see details in Supplementary Note 14). We classified a mutation as explained when the predicted VAF is within a 95% confidence interval (CI) (according to a binomial model with beta prior[34,35]) of the observed VAF (Fig. 6a). When counting the number of explained mutations, we excluded mutations that have low frequency (VAF < 0.2) as well as mutations that are not explained by the copy numbers and proportions inferred by any of the methods. These excluded mutations are more likely to have occurred after CNAs and to be present in smaller subpopulations of cells.

We identified ≈10,600 mutations per prostate cancer sample and ≈9,000 mutations per pancreas cancer sample (Supplementary Fig. 50). We found that for 13/14 patients the copy numbers and proportions inferred by HATCHet yield substantially fewer unexplained mutations (Fig. 6b, c) and lower errors (Supplementary Figs. 51 and 52) than the copy numbers inferred by Battenberg and Control-FREEC, respectively, with the difference on the remaining patient being small. On the prostate cancer dataset, HATCHet explains most of the mutations with high VAF, while the unexplained mutations mostly have lower VAFs (Supplementary Figs. 53 and 54), suggesting that these mutations occurred after the CNAs at the locus, as reported in the published analysis[11]. On the pancreas cancer dataset, we observed that nearly all mutations have low VAFs (Supplementary Fig. 55),

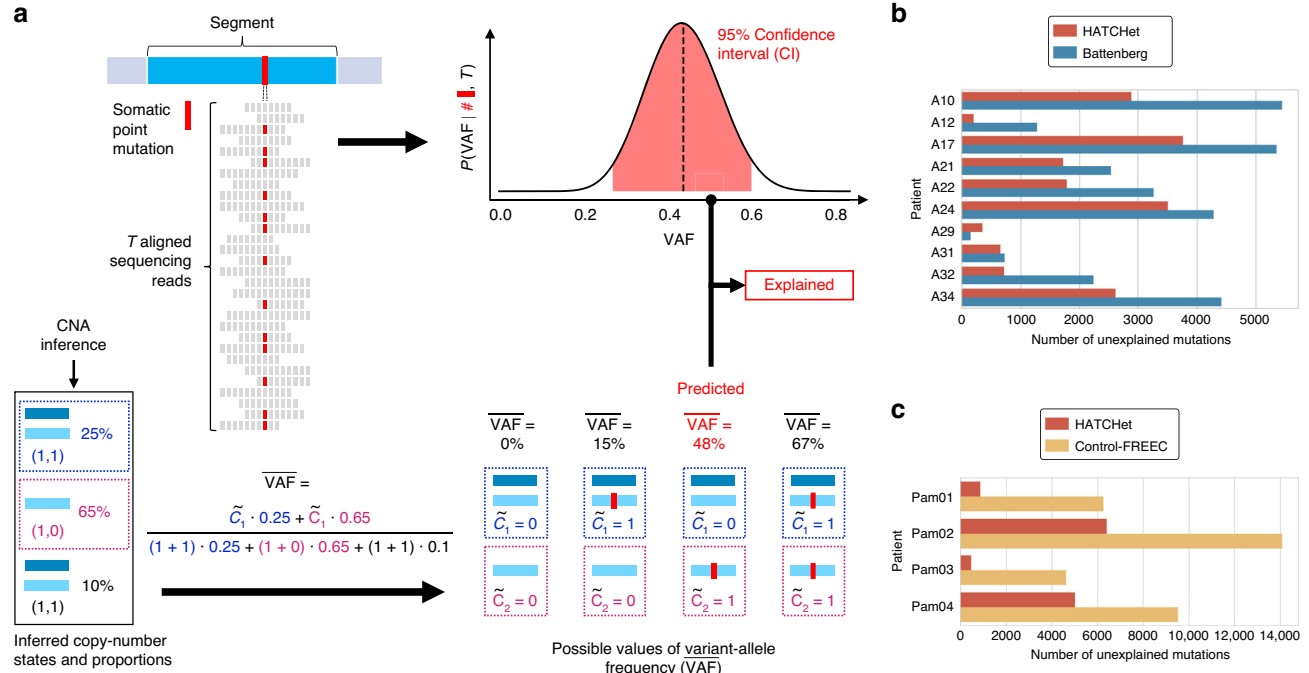

**Fig. 6 HATCHet infers copy-number states and proportions that better explain VAFs of somatic SNVs and small indels. a** A genomic segment (cyan rectangle) harbors a somatic mutation, which corresponds to either a somatic SNV or small indel. Reads with variant allele (red squares) and reference allele (gray squares) are used to estimate the VAF. (Top right) From $T$ sequencing reads (gray rectangles) covering the mutation, a 95% confidence interval (CI, i.e., red area of posterior probability) on the VAF is obtained from a binomial model. (Bottom) Separately, copy-number states and proportions are inferred for this genomic segment. Given the numbers $\tilde{c}_1, \tilde{c}_2$ of mutated copies in each of the two copy-number states, the $\overline{\text{VAF}}$ of the mutation is computed as the fraction of the mutated copies weighted by the proportions of the corresponding copy-number states. Assuming that an allele-specific position is mutated at most once during tumor progression (i.e., no homoplasy), all possible values of $\overline{\text{VAF}}$ are computed according to the possible values of $\tilde{c}_1$ and $\tilde{c}_2$. A mutation is explained if at least one value of $\overline{\text{VAF}}$ is within CI. **b** Over 10,600 mutations identified per prostate cancer patient on average, HATCHet copy numbers (red) yield fewer unexplained mutations than Battenberg (blue) in all patients but A29, where the difference is small. **c** Over 9,000 mutations identified per pancreas cancer patient on average, HATCHet copy numbers yield fewer unexplained mutations in all patients than Control-FREEC.

consistent with low tumor purity as well as the presence of WGDs and/or higher ploidy in these samples. Indeed, SNVs/indels that occur after WGDs alter only one copy of the locus, and thus have low VAF. As lower VAFs are also observed in samples with higher purity (e.g., Pam01_LiM2, Pam01_NoM1, and Pam02_PT18), WGDs and high ploidy are the more likely explanation for the low VAFs, consistent with HATCHet's prediction of WGD in 3/4 patients (Supplementary Fig. 56).

Finally, for each mutation in the prostate cancer patients, we computed the cancer cell fraction (CCF), or fraction of tumor cells that harbor a copy of the mutation, using the method described in Dentro et al.[49] and the copy numbers and proportions inferred by either Battenberg or HATCHet. Across all patients, we found that ≈11% (i.e., ≈200 mutations per patient) of the unexplained mutations that were classified as subclonal (i.e., CCF ≪ 1 and present in a subset of cells) in the published results using Battenberg's copy numbers[11] were explained and classified as clonal (i.e., CCF ≈ 1 and present in all tumor cells) using HATCHet's copy numbers (Fig. 7). For example, in sample A10-E of patient A10 and sample A17-F of patient A17, HATCHet infers clonal CNAs on chromosomes 1p and 8q, respectively, that explain all SNVs at these loci, while Battenberg inferred subclonal CNAs at these loci that result in unexplained SNVs (Fig. 7a, b).

We found a particularly interesting case in two samples A22-J and A22-H of patient A22, where HATCHet explains a large cluster of mutations on chromosome 8p and classifies them as clonal (CCF ≈ 1) while Battenberg does not explain these

mutations and classifies them as subclonal (CCF ≈ 0.4 and CCF ≈ 0.6 in the two samples) (Fig. 7c). This difference is due in part to different copy numbers inferred by the two methods: HATCHet assigned copy-number state (2, 0) to chromosome 8p in both samples A22-H and A22-J, while Battenberg inferred (2, 0) in sample A22-H and (1, 0) in sample A22-J. This demonstrates the advantage of HATCHet's leveraging of information across samples from the same patient. Notably, this cluster of mutations on chromosome 8p was highlighted as a main evidence of polyclonal migration between samples A22-J and A22-H (corresponding to purple cluster in Figure 1 of Gundem et al.[11]), since the unexplained mutations are classified as subclonal in both samples based on Battenberg's results. Based on HATCHet's inferred copy numbers, these mutations are classified as clonal and are not evidence of polyclonal migration.

## Discussion

The increasing availability of DNA sequencing data from multiple tumor samples from the same cancer patient provides the opportunity to improve the copy-number deconvolution of bulk samples into normal and tumor clones. Joint analysis of multiple tumor samples has proved to be of substantial benefit in the analysis of SNVs[11,29,32–36]. However, the advantages of joint analysis have not been exploited in the analysis of CNAs, with all analyses of the prominent multi-sample sequencing datasets[11,12,29,30] relying on CNA methods that analyze individual samples, and in some cases assuming that copy numbers are the same in all tumor cells in a sample.

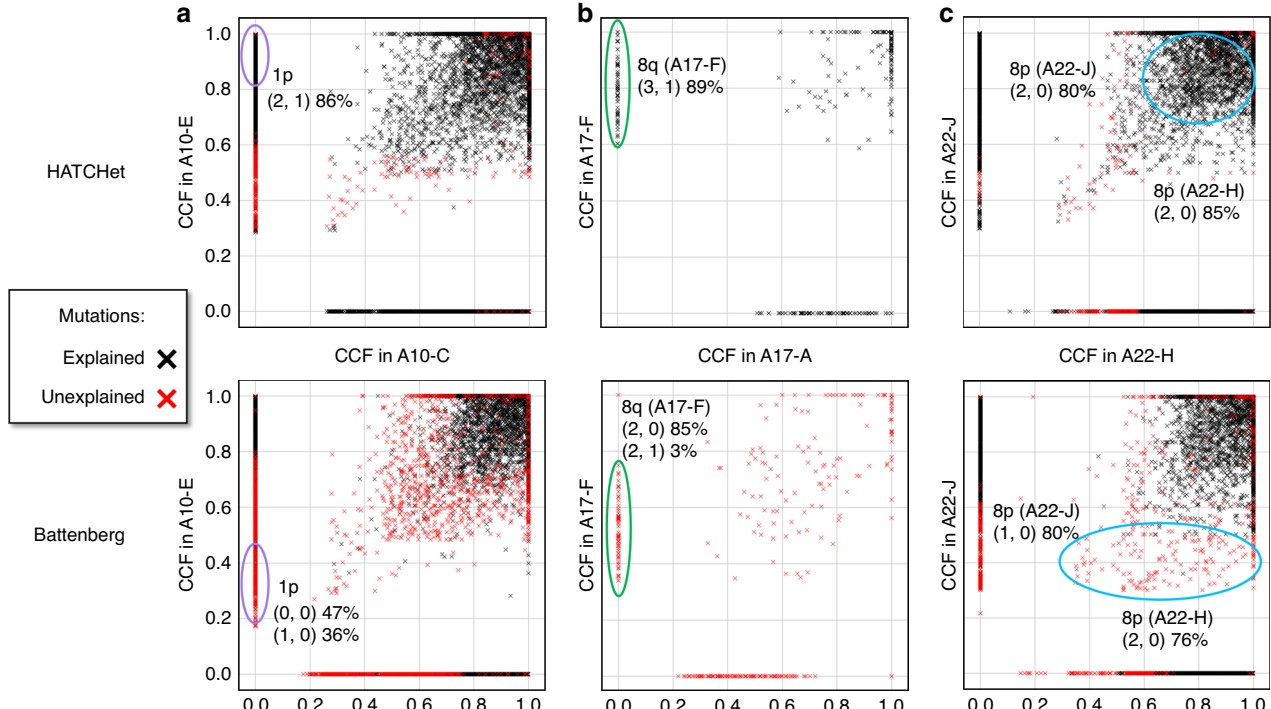

**Fig. 7 HATCHet copy numbers improve estimates of CCFs of somatic mutations in prostate cancer patients. a** CCFs of somatic SNVs and small indels in samples A10-C and A10-E of patient A10 computed from allele-specific copy numbers and proportions inferred by HATCHet (top) and Battenberg (bottom). HATCHet explains a substantial number of mutations that are unexplained by Battenberg; for example, HATCHet infers a clonal CNA on chromosome 1p in A10-E and determines that the mutations at this locus (purple circle) are clonal (i.e., CCF ≈ 1). In contrast, Battenberg infers subclonal CNAs at the same locus, and determines that the same mutations are subclonal (CCF ≈ 0.3). **b** CCFs of somatic SNVs and small indels in samples A17-A and A17-F of patient A17 show groups of mutations that are explained by HATCHet and unexplained by Battenberg (only this subset of mutations is shown here for simplicity). For example, HATCHet infers a clonal CNA on chromosome 8q in A17-F and suggests that mutations in that region (green circle) are clonal (CCF ≈ 1), while Battenberg infers subclonal CNAs and suggests that the same mutations are subclonal (CCF ≈ 0.5). **c** CCFs of somatic SNVs and small indels in samples A22-J and A22-H of patient A22 show a large group of shared mutations on chromosome 8p (cyan circle with CCF > 0 in both samples). HATCHet infers the same copy-number state (2, 0) in both samples, explains these mutations, and suggests that they are clonal. Battenberg infers distinct copy-number states (1, 0) and (2, 0) in the two samples, leaves these mutations unexplained, and suggests that the mutations are subclonal in both samples.

In this paper, we introduced HATCHet, an algorithm to infer allele-specific CNAs and clone proportions jointly across multiple tumor samples from the same patient. HATCHet includes two major enhancements that improve performance over existing methods for copy-number deconvolution. First, we showed that with multiple samples, global clustering of read counts along the genome and across samples becomes an effective strategy, analogous to the clustering of SNVs across samples[32,33,36] but different from the current focus in CNA inference of local segmentation of read counts along the genome. Second, we showed the advantage of separating the two sources of ambiguity in copy-number deconvolution: ambiguity in fractional copy numbers vs. ambiguity in the factorization of fractional copy numbers into integer-valued copy-number states. HATCHet defers the selection of fractional copy numbers, performing model selection in the natural coordinates of copy-number states and clone proportions. We also introduced MASCoTE, a simulator for multi-sample tumor sequencing data that correctly accounts for different genome lengths of tumor clones and WGD. Finally, we showed that HATCHet outperforms existing methods for CNA inference on simulated bulk tumor samples and produces more plausible inferences of subclonal CNAs and WGDs on two cancer datasets.

There are several areas for future improvements. First, while we have shown that HATCHet accurately recovers the major tumor clones distinguished by larger CNAs, HATCHet may miss small CNAs or CNAs at low proportions. One interesting future direction is to perform a second stage of CNA inference using a local segmentation algorithm (e.g., a HMM[17]) informed by the clonal composition inferred by HATCHet. Second, HATCHet's model of RDR and BAF could be improved by modeling additional sources of variation in the data, including replication timing[50] or variable coverage across samples, by considering different generative models for RDR and BAF, and by incorporating additional signals in DNA sequencing reads, such as phasing of germline SNPs[9,27]. Third, HATCHet's modeling of WGD could be further generalized. While recent pan-cancer studies[5–8,12] show that the current assumptions used in HATCHet (namely that a WGD occurs at most once as a clonal event and that additional clonal CNAs also occur) are reasonable for most tumors, HATCHet's model could be extended to allow for multiple WGDs (e.g. hexaploid or higher ploidy), subclonal WGDs, or WGDs occurring without any other clonal CNAs. Fourth, HATCHet's model-selection criterion could be further improved by including additional information such as a more refined model of copy-number evolution[23,24,51–53], and temporal[54] or spatial[13] relationships between clones. Fifth, further improvements integrating CNAs and SNVs are an important future direction. For example, phasing somatic mutations to nearby germline SNPs might provide additional information to identify explained mutations, although in the present study, only a small fraction of the mutations (<0.2% in the prostate and

<0.17% in pancreas cancer patients) are on the same sequencing read as a heterozygous germline SNP. Finally, some of the algorithmic advances in HATCHet can be leveraged in the design of better methods for inferring CNAs and WGDs in single-cell sequencing data.

The increasing availability of DNA sequencing data from multiple bulk tumor samples from the same patient provides the substrate for deeper analyses of tumor evolution over time, across space, and in response to treatment. Algorithms that maximally leverage this data to quantify the genomic aberrations and their differences across samples will be essential in translating this data into actionable insights for cancer patients.

## Methods

**HATCHet algorithm.** We introduce HATCHet, an algorithm to infer allele- and clone-specific CNAs and clone proportions for several tumor clones jointly across multiple bulk tumor samples. We represent the accumulation of all CNAs in all clones by partitioning the $L$ genomic positions of the reference genome into $m$ segments, or clusters, with each segment $s$ consisting of $\ell_s$ genomic positions with the same copy numbers in every clone. Thus, a clone $i$ is represented by a pair of integer vectors $\mathbf{a}_i$ and $\mathbf{b}_i$ whose entries indicate the number of copies of each of the two alleles for each segment. Specifically, we define the copy-number state $(a_{s,i}, b_{s,i})$ of segment $s$ in clone $i$ as the pair of the two integer allele-specific copy numbers $a_{s,i}$ and $b_{s,i}$, whose sum determines the total copy-number $c_{s,i} = a_{s,i} + b_{s,i}$. In addition, we define clone 1 to be the normal (non-cancerous) diploid clone, and thus $(a_{s,1}, b_{s,1}) = (1, 1)$ and $c_{s,1} = 2$ for every segment $s$ of the normal clone. We represent the allele-specific copy numbers of all clones as two $m \times n$ matrices $A = [a_{s,i}]$ and $B = [b_{s,i}]$. Similarly, we represent the total copy numbers of all clones as the $m \times n$ matrix $C = [c_{s,i}] = A + B$. Due to the effects of CNAs, the genome length $L_i = \sum_{s=1}^{m} c_{s,i} \ell_s$ of every tumor clone $i$ is generally different from the genome length $L_1 = 2L$ of the normal clone.

We obtain DNA sequencing data from $k$ samples of a cancer patient and we assume that each tumor sample $p$ is a mixture of at most $n$ clones, with clone proportion $u_{i,p}$ indicating the fraction of cells in $p$ that belong to clone $i$. Note that $0 \leq u_{i,p} \leq 1$ and the sum of clone proportions is equal to 1 in every sample $p$. We say that $i$ is present in $p$ if $u_{i,p} > 0$. The tumor purity $\mu_p = \sum_{i=2}^{n} u_{i,p}$ of sample $p$ is the sum of the proportions of all tumor clones present in $p$. We represent the clone proportions as the $n \times k$ matrix $U = [u_{i,p}]$.

HATCHet starts from the DNA sequencing data obtained from the $k$ samples (Fig. 1a) and infers allele- and clone-specific CNAs in two separate modules. The first module of HATCHet infers the allele-specific fractional copy numbers $f_{s,p}^A = \sum_i a_{s,i} u_{i,p}$ and $f_{s,p}^B = \sum_i b_{s,i} u_{i,p}$ whose sum defines the fractional copy-number $f_{s,p} = f_{s,p}^A + f_{s,p}^B$ (Fig. 1b–d). We represent the allele-specific fractional copy numbers using two $m \times k$ matrices $F^A = [f_{s,p}^A]$ and $F^B = [f_{s,p}^B]$. The second module of HATCHet infers allele- and clone-specific copy numbers $A$, $B$ and clone proportions $U$ by simultaneously factoring $F^A = AU$ and $F^B = BU$ (Fig. 1e, f). Importantly, HATCHet infers two values of $F^A$, $F^B$ according to absence/presence of a WGD, and uses a model-selection criterion to simultaneously choose the number $n$ of clones and the presence/absence of a WGD while performing the copy-number deconvolution (Fig. 1g). We describe the details of these two modules in the next two sections.

**Inference of allele-specific fractional copy numbers.** The first module of HATCHet aims to infer the allele-specific fractional copy numbers $F^A$ and $F^B$ from the DNA sequencing data of $k$ samples. This module has three steps.

The first step of the first module is the computation of RDRs and BAFs (Fig. 1b), which are derived from the DNA sequencing data for every genomic region in each sample. The RDR $r_{s,p}$ of a segment $s$ in sample $p$ is directly proportional to the fractional copy-number $f_{s,p}$. The BAF $\beta_{s,p}$ measures the proportion of the two allele-specific fractional copy numbers $f_{s,p}^A, f_{s,p}^B$ in $F^A$ and $F^B$, respectively. HATCHet computes RDRs and BAFs by partitioning the reference genome into short genomic bins (50 kb in this work) and using the same approach of existing methods[6,9,14–27] to compute appropriate normalizations of sequencing read counts with a matched-normal sample—accounting for GC bias and other biases. Further details are in Supplementary Methods 1 and 2.

The second step of the first module is the inference of the genomic segments that have undergone CNAs directly from the measured RDRs and BAFs. The standard approach to derive such segments is to assume that neighboring genomic loci with similar values of RDR and BAF are likely to have the same copy-number state in a sample. All current methods for CNA identification rely on such local information, and use segmentation approaches, such as Hidden Markov Models (HMMs) or change-point detection, to cluster RDRs and BAFs for neighboring genomic regions[9,14,15,17–19,55–57]. With multiple sequenced samples from the same patient, one can instead take a different approach of identifying segments with the same copy-number state by globally clustering RDRs and BAFs along the entire

genome and simultaneously across multiple samples (Supplementary Fig. 57). Two previous methods, FACETS[18] and CELLULOID[26], clustered segments obtained from a local segmentation algorithm into a small number of distinct copy-number states. HATCHet introduces a global clustering which extends this previous approach in two ways: first, HATCHet jointly analyzes multiple samples from the same patient and, second, HATCHet does not rely on local segmentation.

HATCHet uses a non-parametric Bayesian clustering algorithm[38] to globally cluster the RDRs and BAFs of all genomic bins jointly across all samples (Fig. 1c). Further details are in Supplementary Method 2. Each cluster corresponds to a collection of segments with the same copy-number state in each tumor clone. These clusters are used to define the entries of $F^A$ and $F^B$, playing the role of the segments described above. Although we do not require that clusters contain neighboring genomic loci, we find in practice that our clusters exhibit such locality (see results on cancer datasets). By clustering globally we preserve local information, but the converse does not necessarily hold. The joint clustering across multiple samples is particularly useful in the analysis of samples with low tumor purity. While variations in the values of RDR and BAF cannot be easily distinguished from noise in a single sample with low tumor purity, jointly clustering across samples leverages information from higher purity samples to assist in clustering of lower purity samples (see results on the pancreas cancer dataset).

The last step of the first module is the explicit inference of the allele-specific fractional copy numbers $F^A$ and $F^B$ from the RDRs and BAFs of the previously inferred segments (Fig. 1d). Existing methods[6,9,14–22,25–27]—including widely used methods such as ABSOLUTE[6], ASCAT[14], Battenberg[9], TITAN[17], cloneHD[25]—do not attempt to directly infer fractional copy numbers, but rather attempt to fit other variables, specifically the tumor ploidy and tumor purity (or equivalent variables as the haploid coverage, Supplementary Method 1). However, the values of these variables are difficult to infer[21,22,25,27] and often require manual selection[6,7,12,27]. Further details regarding tumor purity and tumor ploidy are reported below in the comparison of HATCHet and existing methods.

We introduce an approach to estimate $F^A$ and $F^B$ with rigorous and clearly-stated assumptions. First, in the case without a WGD, we assume there is a reasonable number of genomic positions in segments whose total copy number is 2 in all clones; this is generally true if a reasonable proportion of the genome is not affected by CNAs and hence diploid. Second, in the case where a WGD occurs, we assume two groups of segments whose total copy numbers are distinct and the same in all clones; this is also reasonable if some segments are affected only by WGD and tumor clones accumulate clonal CNAs during tumor evolution. More specifically, we scale the RDR $r_{s,p}$ of each segment $s$ in every sample $p$ into the fractional copy number $f_{s,p}$ and separate $f_{s,p}$ into the allele-specific fractional copy numbers $f_{s,p}^A, f_{s,p}^B$ using the BAF $\beta_{s,p}$. The following theorem states that the assumptions above are sufficient for scaling RDRs to fractional copy numbers.

*Theorem 1*: The fractional copy number $f_{s,p}$ of each segment $s$ in each sample $p$ can be derived uniquely from the RDR $r_{s,p}$ and either (1) a diploid clonal segment $s'$ with total copy number $c_{s',i} = 2$ in every clone $i$ or (2) two clonal segments $s'$ and $z'$ with total copy numbers $c_{s',i} = \omega_{s'}$ and $c_{z',i} = \omega_{z'}$ for all tumor clones $i$, and such that $r_{s',p}(\omega_{z'} - 2) \neq r_{z',p}(\omega_{s'} - 2)$ for all samples $p$.

Notably, this theorem states that the scaling is independent of other copy numbers in $A$, $B$, and $C$ as well as the clone proportions in $U$.

To apply this theorem, HATCHet employs a heuristic to identify the required segments and their total copy numbers; this heuristic leverages the RDRs and BAFs jointly across all samples. First, in the case of no WGD, we aim to identify diploid segments with a copy-number state $(1, 1)$. These segments are straightforward to identify: first, diploid segments will have $\beta_{s,p} \approx 0.5$ in all samples $p$; and second we assume that a reasonable proportion of the genome in all samples will be unaffected by CNAs and thus have state $(1, 1)$. As such, we identify the largest cluster of segments with $\beta_{s,p} \approx 0.5$ in all samples $p$, and apply Theorem 1. Second, in the case of a WGD, we assume that at most one WGD occurs and that the WGD affects all tumor clones. These assumptions are consistent with previous pan-cancer studies of WGDs[5–8,12]. Under these assumptions, the segments $s$ with $\beta_{s,p} \approx 0.5$ have copy-number state $(2, 2)$, as a WGD doubles all copy numbers. Thus, we use the second condition of Theorem 1 and aim to find another group of segments with the same state in all tumor clones. More specifically, HATCHet finds segments whose RDRs and BAFs in all samples indicate copy-number states that result from single-copy amplifications or deletions occurring before or after a WGD[5]; for example, copy-number state $(2, 0)$ is associated to a deletion occurring before a WGD while copy-number state $(2, 1)$ is associated to a deletion occurring after a WGD. Moreover, we select only those groups of segments whose RDRs and BAFs relative to other segments are preserved in all samples; such preservation indicates that the copy-number state is fixed in all tumor clones (Fig. 1f). Further descriptions of Theorem 1 and this heuristic are in Supplementary Method 3.

**Inferring allele- and clone-specific copy numbers and clone proportions.** The second module of HATCHet aims to derive allele- and clone-specific copy numbers $A$, $B$ and clone proportions $U$ from the two values of allele-specific fractional copy numbers $F^A$ and $F^B$ that were estimated in the first module. The second module has two steps.

The first step of the second module is the inference of $A$, $B$, and $U$ from each estimated value of $F^A$ and $F^B$ (Fig. 1e). Since the samples from the same patient are

related by the same evolutionary process, we model the fractional copy numbers jointly across the $k$ samples such that $F^A = AU$ and $F^B = BU$. As such, the problem that we face is to simultaneously factorize $F^A$ and $F^B$ into the corresponding allele-specific copy numbers $A$, $B$ and clone proportions $U$ for some number $n$ of clones. Formally, we have the following problem.

*Problem 1*: (Allele-specific Copy-number Factorization (ACF) problem) Given the allele-specific fractional copy numbers $F^A$ and $F^B$ and the number $n$ of clones, find allele-specific copy numbers $A = [a_{s,i}]$, $B = [b_{s,i}]$ and clone proportions $U = [u_{i,p}]$ such that $F^A = AU$ and $F^B = BU$.

While the ACF problem is a mathematically elegant description of the copy-number deconvolution problem, there are two main practical issues: first, measurement errors in $F^A$ and $F^B$ may result in the ACF problem having no solution, and second the ACF problem is an underdetermined problem and multiple factorizations of a given $F^A$ and $F^B$ may exist. To address the first issue, we do not solve the simultaneous factorization $F^A = AU$ and $F^B = BU$ exactly, but rather minimize the distance between the estimated fractional copy numbers $F^A$ and $F^B$ and the factorizations $AU$ and $BU$, respectively, weighted by the corresponding size of the clusters. In particular, we define the distance $\| F^A - AU \| = \sum_{s=1}^{m} \sum_{p=1}^{k} \ell_s |F_{s,p}^A - \sum_{1 \le i \le n} a_{s,i} u_{i,p}|$, where $\ell_s$ is the genomic length of the cluster $s$. We also define the corresponding distance for $F^B$, $B$, and $U$.

To address the second issue of an underdetermined system, we impose three additional and reasonable constraints. All of these constraints are optional and user-selectable. First, since we do not expect copy numbers to be arbitrarily high—especially for large genomic regions—we assume that the total copy numbers are at most a value $c_{\max}$. Second, to avoid overfitting errors in fractional copy numbers by clones with low proportions, we require a minimum clone proportion $u_{\min}$ for every tumor clone present in any sample. Third, we impose an evolutionary relationship between the tumor clones requiring that each allele of every segment $s$ cannot be simultaneously amplified and deleted in distinct clones; i.e., either $a_{s,i} \ge \theta$ or $a_{s,i} \le \theta$ for all clones $i$, where $\theta = 1$ when there is no WGD and $\theta = 2$ when there is a WGD. The same constraint also holds for $b_{s,i}$. These constraints improve the solutions to the copy-number deconvolution problem[23,24] and are less restrictive than the ones usually applied in current methods which, for example, assume that: tumor clones have at most two copy-number states $(a_{s,i}, b_{s,i})$, $(a_{s,j}, b_{s,j})$ per segment and the difference between allele-specific copy numbers is at most 1[9,19], i.e., $|a_{s,i} - a_{s,j}| \le 1$ and $|b_{s,i} - b_{s,j}| \le 1$; or all clones have either a diploid copy-number state $(1, 1)$ or a unique aberrant state $(a, b) \ne (1, 1)$ in every cluster $s$[17,18]; or every tumor clone $i$ has either $c_{s,i} \ge 2$ or $c_{s,i} \le 2$ for every cluster $s$[21,22]; or there always exist segments with total copy number equal to 2[18,25]. We thus have the following problem.

*Problem 2*: (Distance-based Constrained Allele-specific Copy-number Factorization (D-CACF) problem) Given the allele-specific fractional copy numbers $F^A$ and $F^B$, a number $n$ of clones, a maximum total copy number $c_{\max}$, a minimum clone proportion $u_{\min}$, and a constant value $\theta \in \{1, 2\}$, find allele-specific copy numbers $A = [a_{s,i}]$, $B = [b_{s,i}]$ and clone proportions $U = [u_{i,p}]$ such that: the distance $D = \| F^A - AU \| + \| F^B - BU \|$ is minimum; $a_{s,i} + b_{s,i} \le c_{\max}$ for every cluster $s$ and clone $i$; either $u_{i,p} \ge u_{\min}$ or $u_{i,p} = 0$ for every clone $i$ and sample $p$; for every cluster $s$, either $a_{s,i} \ge \theta$ or $a_{s,i} \le \theta$ for all clones $i$; for every cluster $s$, either $b_{s,i} \ge \theta$ or $b_{s,i} \le \theta$ for all clones $i$.

We design a coordinate-descent algorithm[23,24] to solve this problem by separating the inference of $A$, $B$ from the inference of $U$ and iterating these two steps until convergence for multiple random restarts. We also derive an ILP formulation that gives exact solutions for small instances. HATCHet uses one of these two algorithms to infer $A$, $B$, $U$ from $F^A$, $F^B$. Further details of this problem and methods are in Supplementary Method 4.

Finally, the last step of the second module uses a model-selection criterion to joint select the number $n$ of clones and the occurrence of a WGD (Fig. 1f). Model selection is essential because variations in the fractional copy numbers $F^A$ and $F^B$ can be fit by increasing the total number $n$ of clones, increasing the number of clones present in a sample, or introducing additional copy-number states in a sample by inferring subclonal CNAs or WGD. There is a trade-off between these options. For example, a collection of clusters that exhibit many different copy-number states may be explained in different ways: e.g., one could increase $n$ and mark some clusters as subclonal, or one could infer the presence of a WGD which will increase the number of clonal copy-number states (Supplementary Fig. 1). Existing methods either: do not perform model selection and assume that the number $n$ of clones is known[21,22,26,27]; consider segments independently[6,9,17–19] (Supplementary Fig. 28), perhaps increasing the sensitivity to detect small subclonal CNAs, but with a danger of overfitting the data and overestimating $n$ and the presence of subclonal CNAs; ignore the trade-off between subclonal CNAs (related to a higher number of clones) and WGD (related to a higher value of tumor ploidy) by not evaluating the presence or absence of WGD in the model selection. Following the factorizations of the two values of $F^A$ and $F^B$ (corresponding to the cases of WGD and no WGD), HATCHet chooses the simplest solution that minimizes the total number $n$ of clones across all samples. Further details on the model-selection procedure are in Supplementary Method 5.

**Comparison of HATCHet and existing methods for copy-number deconvolution.** We summarize some of the main differences between HATCHet and existing methods for copy-number deconvolution (see also Supplementary Table 1). First, HATCHet models allele-specific copy numbers, while many methods do not[20–24].

Second, HATCHet models dependencies between segments as clones, while most of the widely used methods[6,9,14–19] analyze each segment independently and discard the global dependency between segments (Supplementary Fig. 28). Third, HATCHet models dependencies between samples and uses a global clustering approach to infer segments jointly across samples. In contrast, existing methods[6,9,14–22,26,27] analyze samples independently and do not preserve clonal structure across samples; there is one exception which is cloneHD[25], but cloneHD infers segments from each sample independently, and also assumes that every sample comprises the same set of few (2–3) clones and is thus not suitable to analyze samples comprising distinct clones[25].

Finally, HATCHet introduces an explicit model-selection criterion to select among different allele- and clone-specific copy numbers and clone proportions that explain the observed DNA sequencing data. There are often multiple possible mixtures of allele-specific copy numbers that explain the measured RDRs and BAFs: for example, segments with distinct values of RDR and BAF could be explained as either subclonal CNAs or clonal CNAs with high copy numbers, e.g., due to a WGD. It is difficult to distinguish these cases because the total length of the genome of each tumor clone is unknown. HATCHet introduces a model-selection criterion which separates two distinct sources of this ambiguity: (1) the inference of allele-specific fractional copy numbers $F^A$ and $F^B$, which are not uniquely determined by the measured RDRs and BAFs; (2) the inference of the allele- and clone-specific copy numbers $A$, $B$ and the clone proportions $U$. Importantly, HATCHet evaluates two possible values of $F^A$, $F^B$—corresponding to the occurrence of WGD or not—and defers the selection of a solution until after the copy-number deconvolution. Thus, HATCHet performs model selection in the natural coordinates of the problem, i.e., $A$, $B$, and $U$, and evaluates the trade-off between inferring subclonal CNAs (and thus more clones in a sample) or a WGD (Supplementary Fig. 1), when modeling a large number of distinct copy-number states. Supplementary Table 2 lists the parameters used in HATCHet's model-selection criterion and Supplementary Table 3 provides the default values of these parameters.

In contrast, existing methods[6,9,14–22,25–27] for copy-number deconvolution do not distinguish different solutions using the variables $A$, $B$, and $U$, but rather use the variables tumor purity $\mu_p = 1 - u_{1,p}$ and tumor ploidy $\rho_p = \frac{1}{\mu_p} \frac{\sum_{i=2}^{n} u_{i,p} L_i}{L}$ (or equivalent variables such as the haploid coverage, Supplementary Method 1). However, tumor purity $\mu_p$ and tumor ploidy $\rho_p$ are composite variables that sum the contributions of the unknown integer copy numbers $A$, $B$ and the proportions $U$ of multiple clones in a sample. Because of their composite nature, tumor purity and tumor ploidy are both difficult to infer[21,22,25,27] and not ideal coordinates to evaluate tumor mixtures. This is because multiple values of tumor purity $\mu_p$ and tumor ploidy $\rho_p$ may be equally plausible for the same values of RDR and BAF, particularly when more than one tumor clone is present or when a WGD occurs (Supplementary Figs. 2 and 3). Not surprisingly, existing methods that rely on tumor purity and ploidy typically require manual inspection of the output to evaluate the presence of WGD[6,7,12,27]; the few methods that automate the prediction of WGD are based on biased criteria or unstated, restrictive assumptions[9,17,25].

We note that HATCHet does not directly reconstruct a tumor phylogenetic tree. However, the copy numbers inferred by HATCHet can be used as input to methods for phylogenetic reconstruction. For example, the integer copy numbers inferred by HATCHet can be input to MEDICC[51] or CNT[52,53], and the fractional copy numbers can be input to CNT-MD[23,24] or Canopy[37].

**Simulating bulk tumor sequencing data with MASCoTE.** We introduce MAS-CoTE, a method to simulate DNA sequencing data from multiple bulk tumor samples that correctly accounts for tumor clones with varying genome lengths. The simulation of DNA sequencing data from bulk tumor samples that contain large-scale CNAs is not straightforward, and subtle mistakes are common in previous studies. Suppose $R$ sequencing reads are obtained from a sample consisting of $n$ clones with clone proportions $u_1, \ldots, u_n$. Assuming that reads are uniformly sequenced along the genome and across all cells, what is the expected proportion $v_i$ of reads that originated from clone $i$? Most current studies[15–17,25,39–44] that simulate sequencing reads from mixed samples compute $v_i$ as a function of $u_i$ without taking into account the corresponding genome length $L_i$. For example, Ha et al.[17] and Adalsteinsson et al.[39] artificially form a mixed sample of two clones by mixing reads from two other given samples in proportions $v_i = \frac{u_i}{\tilde{u}_i}$ where $\tilde{u}_i$ is the clone proportion of the single tumor clone $i$ uniquely present in a given sample. Another example is Salcedo et al.[42] that simulates the reads for each segment $s$ separately by setting $v_{s,i} = \ell_s \frac{c_{s,i} u_i}{M}$ for every clone $i$ where $M = \max_s f_s$ is the maximum fractional copy number. However, such values of $v_i$ are the correct proportions only when the genome lengths of all clones are equal, e.g., $L_i = 2L$ for every clone $i$. Using an incorrect proportion $v_i$ leads to incorrect simulations of read counts, particularly in samples containing WGDs or multiple large-scale CNAs in different clones (Supplementary Figs. 4 and 5). In fact, read counts depend on the genome lengths of all clones in the sample[58] and the correct proportion $v_i = \frac{u_i L_i}{\sum_{j=1}^{n} u_j L_j}$ is equal to the fraction of genome content in a sample

belonging to the cells of clone $i$. Moreover, the expected proportion $v_{s,i}$ of reads in segment $s$ that originate from clone $i$ is equal to $v_{s,i} = \ell_s \frac{c_{s,i} u_i}{\sum_{j=1}^{n} u_j L_j}$, the fraction of the genome content from segment $s$ belonging to the cells of clone $i$ (Supplementary Method 1).

To address these issues, we develop MASCoTE to correctly simulate DNA sequencing reads of multiple mixed samples obtained from the same patient (Supplementary Fig. 6). MASCoTE simulates the genomes of a normal clone and $n - 1$ tumor clones, which accumulate CNAs and WGDs during tumor evolution; these clones are related via a phylogenetic tree. As such, every sample comprises a subset of these clones and the corresponding sequencing reads are simulated according to the genome lengths and proportions of the clones. More specifically, MASCoTE is composed of four steps: (1) MASCoTE simulates a diploid haplotype-specific germline genome (Supplementary Fig. 6a); (2) MASCoTE simulates the genomes of $n - 1$ tumor clones that acquire different kinds of CNAs and WGDs—according to the distributions in size and quantity reported in previous pan-cancer studies[5]—in random order through a random phylogenetic tree (Supplementary Fig. 6b); (3) MASCoTE simulates the sequencing reads from the genome of each clone through standard methods[59] (Supplementary Fig. 6c); (4) MASCoTE simulates each sample $p$ by considering an arbitrary subset of the clones (always containing the normal clone) with random clone proportions and by mixing the corresponding reads using the read proportion $v_{i,p} = \frac{u_{i,p} L_i}{\sum_{1 \le j \le n} u_{j,p} L_j}$ (Supplementary Fig. 6d). Further details about this procedure are in Supplementary Method 6.

**Bioinformatics analysis**. We applied MASCoTE with default values of all parameters to simulate DNA sequencing reads for 64 patients, half with a WGD and half without a WGD. For each patient, we simulated 3–5 bulk tumor samples, with a total of 256 samples across all 64 patients. For each patient, we simulated 2–4 clones (including the normal clone) with CNAs of varying size using the relative frequencies reported in pan-cancer analysis[5] and including: focal CNAs < 1Mb, small CNAs between 3 and 5 Mb, medium CNAs between 10 and 20 Mb, and chromosome arm and whole chromosome CNAs. We provided the human reference genome hg19 and the database dbSNP of known SNPs[60] to MASCoTE for generating a haplotype-specific genome for each normal clone. We ran every method on the simulated samples by using the default available pipelines. Details about the experimental setting of every method are described in Supplementary Note 1.

We applied HATCHet to analyze 49 samples from 10 prostate cancer patients in Gundem et al.[11] and 35 samples from four pancreas cancer patients in Makohon et al.[30] using the published BAM files. In addition to one or more BAM files from the same patient, HATCHet requires two other sources of information: a matched-normal sample and the reference genome used to align the sequencing reads. We used the available matched-normal sample for every patient and the reference genome corresponding to the alignments in the BAM files, i.e., GRCh37 for the prostate cancer patients and hg19 for the pancreas cancer patients. HATCHet used BCFtools (v1.7)[61] to identify germline heterozygous SNPs with the provided matched-normal sample and reference genome. For each patient, we applied HATCHet on all the corresponding samples using the default values of all parameters: genomic bin size of 50 kb, maximum total copy-number $c_{max} = 12$, and minimum clone proportion $u_{min} = 0.03$ (for patients A22, A21, Pam03, and Pam04, we used $u_{min} = 0.15$ since these patients exhibited high variance in RDRs and BAFs). Further details are in Supplementary Note 6. Lastly, we used Varscan 2 (v2.3.9)[62] with default parameters and filters to identify somatic SNVs and small indels.

**Reporting summary**. Further information on research design is available in the Nature Research Reporting Summary linked to this article.

## Data availability

Whole-genome DNA sequencing data for the prostate and pancreas cancer datasets analyzed in this study are available from the European Genome-phenome Archive (EGA) under accession numbers EGAS00001000262 and EGAS00001002186, respectively. Whole-exome DNA sequencing data for breast cancer patients in Kim et al.[45] and Casasent et al.[46] are available from the NCBI Sequence Read Archive (SRA) under accession numbers SRP114962 and SRP116771. All the processed simulated data, the results of all methods on simulated data, and the results of HATCHet on the prostate and pancreas cancer datasets are available on GitHub from https://github.com/raphael-group/hatchet-paper and on Zenodo from https://doi.org/10.5281/zenodo.3830088.

## Code availability

HATCHet is available on GitHub at https://github.com/raphael-group/hatchet. MASCoTE is available on GitHub at https://github.com/raphael-group/mascote.

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

## Acknowledgements

We thank Christine Iacobuzio-Donahue and Alvin Makohon-Moore for assistance in obtaining the copy-number data from their publication[30]. We thank Stefan Dentro, Peter Van Loo, and David Wedge for assistance in running Battenberg on our simulated data. We thank Gavin Ha for assistance in running TITAN on our simulated data. This work was supported by a US National Institutes of Health (NIH) grants R01HG007069 and U24CA211000 and US National Science Foundation (NSF) CAREER Award (CCF-1053753) to B.J.R.

## Author contributions
S.Z. and B.J.R. conceived the project, developed the theory and algorithms, and wrote the paper; S.Z. implemented the algorithms and performed the analyses.

## Competing interests
B.J.R. is a cofounder of, and consultant to, Medley Genomics. S.Z. declares no competing interests.
