## [Peer Review File · Nature Communications]

REVIEWER COMMENTS

Reviewer #3 (Remarks to the Author):

We thank the authors for their detailed responses to our initial comments--the authors have clarified most of the concerns by adding more comparisons between known software tools (including Canopy, Battenberg and ReMixT), as well as using somatic SNVs and indels to support their findings. Many of the provided explanations are much clearer. In particular, the author responses helped highlight what HATCHET and MASCOTE do differently compared to other methods, so the novelty of the methods is more apparent. Table R1 is a great addition to clarify not only how HATCHet differs from existing methods but also how to conceptually call SCNAs from bulk WGS data (particularly for readers relatively new to the field).

The authors do concede that cloneHD has implemented joint-sample calling to detect CNAs, but they stress that HATCHet is one of the few methods that carries out joint-sample calling and to better effect than cloneHD (with advantages over other methods). Based on the revision and response, it appears that the chief innovations of HATCHet are (1) to determine which segments of the genome share the same copy number state on the basis of a clustering algorithm applied to only BAF and RDR (regardless of WGD) and (2) to determine whether WGD occurred until after estimating allele-specific fractional copy numbers and clonal fractions in order to better handle the uncertainty of calling CNAs in samples with possible WGDs -- which does represent a useful insight. Otherwise, the revised manuscript is much clearer, particularly the methods section which describes HATCHet's mathematical logic and design choices more plainly. The manuscript as presented may be suitable for publication, but we have the following comments for the authors to consider.

Major comments

1. Figure 6 describes using HATCHet to explain somatic SNV allele frequencies after CNAs and WGD events. However, there could be a number of somatic SNVs that occurred after the CNAs. While the authors treated mutations as "explained" if only when their VAFs were within 95% CI of predicted VAFs from mutations occur before the CNAs, a large proportion of somatic SNVs could arise later than the CNAs and would have much lower VAFs highly deviated from their expectations. As a result, these SNVs could not serve as a good dataset to compare the performance of different methods. Is it possible to use read-level phasing information of allele-specific CNAs and the linked somatic variants to do a better evaluation? An analysis of heterozygous polymorphisms within regions affected by CNAs would provide a more convincing benchmark -- the true minor-allele fraction of hSNPs within genomic duplication should be inversely proportional to the estimated CN of the region. Additionally, any somatic SNVs phased to a nearby hSNP should show similar results as the hSNP in the proposed simulation.

Minor comments

1. In Figure 3E, why did HATCHet call the small "cloud" to the left of (1,1) as "clonal"? It looks very subclonal (and I'm guessing it corresponds to the dip near chr11). The cloud could represent a sequencing artefact, given how the (2,1) region nearby also appears to dip. Additionally, it seems strange that Battenberg would deem the (2,1) cluster of bins as a subclonal region when it had no such problem in the B/C. Is Battenberg being thrown off by coverage biases?

2. For figures 4 and 5, it would help to show which of the clusters (in panels B onward) were called in the published analysis and what were the estimated copy number states for the clusters that HATCHet did deem as subclonal or arising from a WGD. As presented, figures 4 and 5 are not clear enough to visually show what unique information about SCNAs would HATCHet provide, so some additional annotation would be helpful

Reviewer #4 (Remarks to the Author):

HATCHet, a method for studying tumor clonal architecture from multi-sample bulk sequencing data and using copy number aberrations as identifiers of individual tumor clones, is introduced. Similarly to many existing methods, in particular those designed for SNVs, HATCHet uses all samples simultaneously and assuming that each bulk tumor sample is a mixture of distinct clones (I will refer to each population of cells with unique copy number profile as clone). The set of clones typically consists of a few clones (in the simulated data up to 3 tumor clones), with one of them representing population of healthy cells, whereas the others are tumor clones characterized (defined) with unknown copy number profiles. Clones are assumed to be shared among bulk samples, with varying proportions, which is a standard assumption well motivated by tree of tumor evolution that is common to all samples. It is allowed that some clones are absent from a given sample, but if a clone is present then at least u_{\min} fraction of cells in the sample are required to belong to this clone, where u_{\min} is a parameter. In addition to inferring copy number profiles of clones, method also infers the number of clones and their prevalence in tumor samples. Two distinct cases are considered when inferring the unknowns:

1. case where it is assumed that no whole genome duplication (WGD) occurred
2. case where it is assumed that WGD occurred

Using selection criterion it is decided which of the two is more likely and the number of clones is selected as well. HATCHet is not limited to working with the total copy number, but distinguishes between minor and major alleles by the use of standard B-allele frequencies computed at the sites of heterozygous SNPs.

Looking into the method, genome is first segmented into small bins and read-depth ratio (RDR) and B-allele frequency (BAF) values are computed for each bin in each sample. These values are then combined in a specific order in vector of dimension $2k$, where k is the number of bulk samples (each sample gives one RDR and one BAF value) and clustered through BPNY (reference provided in the Supp Material). During this clustering, bins are clustered into m clusters, with number of clusters inferred by the clustering algorithm. Each cluster consists of a set of genomic bins sharing similar BAF and RDR values across all samples. Note that the inferred clusters do not necessarily contain only block of adjacent bins but, for example, bins from different chromosomes can be clustered together as soon as their BAF and RDR values are similar across all samples. The algorithm then mostly works with the obtained clusters instead of bins. Assuming correct clustering, RDR and BAF estimates of clusters are expected to be more stable than these of individual bins. In addition, clustering also leads to significant dimensionality reduction as the number of clusters is usually small

in comparison to the number of bins.

After that, fractional copy numbers are estimated. This step requires sequencing data of matching normal (healthy) tissue and assumes that a good fraction of the genome is diploid and not affected by copy number aberrations if no WGD is present (similar assumption is made in case where WGD is present). The fractional copy number values are then used as the input to another important part of the algorithm where allele-specific copy numbers of all segments are inferred together with clonal frequencies. This inference is done using, from the algorithmic point of view, minor modifications of previously published ideas and solutions based on coordinate-descent algorithm for some of the problems presented in "Phylogenetic Copy-Number Factorization of Multiple Tumor Samples" (by the same authors, published in Journal of Computational Biology). Note that this was later also used in "Deconvolution and phylogeny inference of structural variations in tumor genomic samples" (Eaton et al., Bioinformatics, 2018). In addition to the coordinate-descent algorithm, the optimization problem, which is not solvable by most (if not all) of the available solvers in the introduced form, is transformed to an instance of classic ILP using standard linearization techniques.

Assume, for simplicity of notation, that in this paragraph of the review $u = u_{ip}$ and $x = x_{ip}$. For $u_{min} > 0$ (which is a case, otherwise having u_{min} is meaningless), combination of constraints (46) and (47) given in the Supplementary Material would force that all x 's are set to 1. Namely, constraint (46) can be rewritten as

$$u \geq u_{min} + (1-x)$$

so setting variable x to 0 would imply that $u \geq 1 + u_{min} > 1$, which is impossible due to the defined range of variables u (as well as due to the constraint (45)). Alternatively, $x=0$ would enforce that $u=0$ through (47) and then (46) will not be fulfilled. Therefore x would never be set to 0 enforcing that each clone is present in each sample that is obviously unrealistic requirement.

The correct system is:

$$u + 1 - x \geq u_{min}$$

$$u \leq x$$

Alternatively, one can impose the following constraints:

$$u \geq u_{min} * x$$

$$u \leq x$$

I hope that this is just one of numerous typos in the manuscript (see below). It is definitely very bad place to have typos as this would most likely waste hours of time of the potential user interested in re-implementing this part of the tool (such re-implementation might be needed if, for example, user does not have Gurobi license and wants to re-implement this in other optimization software, like CPLEX or some freely available ILP solver). If this is not a typo, the authors might need to re-run the experiments performed.

While considering the possibility of existence of whole genome duplications (WGDs), three main assumptions about WGDs are made: (i) WGDs occur at most once (ii) if WGD occurs, then it affects all tumor clones (iii) if WGD is present, there also exist some other clonal copy number aberrations. While each of the assumptions is not necessarily always true in practice, I find the first two of them

quite fair. For example, there is indeed evidence in the literature that WGDs are early events in carcinogenesis. The third assumption is questionable as single WGD combined with SNVs, indels, fusions and/or translocations can suffice for cancer formation. Although future studies, primarily those involving single-cell data, will provide better insight on this, I am of the opinion that it is more likely that this assumption is true than not in most cases, but I recommend adding very brief discussion on the potential limitation of the method due to this assumption.

Why the inference of similar fractions of the genome affected by CNAs implies that subclonal CNAs inferred by Battenberg are clonal (see sentence "Second, both methods inferred similar fractions of the genome affected by CNAs on these 20 samples (Fig. S34A), suggesting that the subclonal CNAs inferred by Battenberg are clonal instead")?

The next sentence, "Finally, we found that ReMixT's inference of subclonal CNAs from the same dataset was more similar to HATCHet than Battenberg". What is the main message that one should take from this sentence?

Why is comparison of results of HATCHet and Control-FREEC on pancreatic cancer dataset relevant? Can available alternative methods give the same insight as HATCHet into clonal composition of these tumors?

Looking into Figure S1, how does blue subclone differ from the healthy cells? I see blue profiles twice, each time it is (1,1). Furthermore, why some of the clusters have black and blue profiles associated with them, whereas the others have only black? Overall, I do not find explanation (caption) of this figure clear enough.

The end of Section 2.4: Why is having one of WGD and subclonal CNAs, but not both of these, a good indication of the quality of solution? Can't WGD introduce higher genomic instability leading to the subsequent subclonal CNAs?

Overall, while the results on real sequencing data to some extent demonstrate that considering multiple samples simultaneously might provide some advantage to clonal architecture reconstruction method, many of the results are anecdotal, debatable and it is not very convincing why solutions reported by HATCHet are a good explanation of the observed read counts. Instead of comparing HATCHet's results with the results of available alternatives on datasets for which even no proxy of ground truth is available, have the authors considered analyzing copy number profiles of tumors for which copy number profiling at single-cell level was performed? A lot of work on this was done for example by Nicholas Navin, one of the pioneers of single-cell sequencing, and his group at MD Anderson Cancer Center. There are also many studies from the other groups and growing number of datasets with available single-cell and matching bulk sequencing data.

I also have many other comments and suggestions related to the manuscript and the repository and they are listed below.

Isn't the variable forming the left hand side of Supp Equation (99) already explicitly defined in the few lines above as a variant allele frequency of SNV e in sample p ? I think that the current presentation is very confusing. On one side of Supp Equation (99) we have a very clearly defined VAF value, which is given as a function of variant and total read counts that are directly observed. Then, this value is expressed as a function of variables f , c and u through Supp Equations (99) and (100), where u 's are unknowns and f and c are defined at the level of cluster of segments (and their values depend on set of reads spanning many different genomic coordinates).

In Supp Equation (101), why $f_{\{s,p\}}$ is used in denominator? In other words, why elements of matrix F are used instead of elements of the product $(A+B)U$ of inferred matrices?

On page 16, in sentence "We also define the corresponding distance for $F^A B$, B , and b .", what is b ? Should this be U ?

How do the authors justify use of minimum subclonal prevalence as high as 15% for several patients analyzed in real data section? How should a user set this important parameter?

In Supp Equation (1) denominator seems to be incorrect. Similarly, I think that the left hand side in Supp Equation (7) has a typo ($b \rightarrow s$).

What is the purpose of adding Supp Equation (11)?

In Supp Equation (17), undefined variables are used in the nominator ($\hat{u}_{\{i,p\}}$, in LaTeX notation).

In Figure S11, part A, right part (Copy-number profiles), for segment s_5 , why is it composed of 70% of total copy number 3 and 50% of total copy number 4 when 70% of (2, 1) and 20% of (3, 1) are shown in the left.

In the introduction in Supp Section B.2., there is an incomplete part of the sentence: "i.e. a cluster s is tumor-clonal if $(a_{\{s,2\}}, b_{\{s,2\}}), (a_{\{s,3\}}, b_{\{s,3\}}), \dots, (a_{\{s,n\}}, b_{\{s,n\}})$."

In the sentence introducing Supp Equation (24), γ_p , which does not appear in the system, is mentioned.

Please remind a reader of Supp Equation (8) and dependency of $f_{\{s,p\}}$, γ_p and $r_{\{s,p\}}$. All in all, I recommend expanding $f_{\{s,p\}}$ based on Equation (8) and discussing what are known (observed) and unknown variables. When it comes to solving the system, it is obviously very trivial and

straightforward task and full details of solving it can be omitted (as is already the case). Furthermore, the authors talk about unknown μ_p (before Supp Equation (24)) and then after Supp Equation (24) they provide formula for τ_p , yet another mistake.

In the sentence starting on 5-th line on supp page 36, sentence should end with $\binom{t}{b}$, not $\binom{b}{b}$?

At the very top of supp page 42, $|a_{s,i} - a_{s,j}| \leq 1$ appears twice. The second one should be $|b_{s,i} - b_{s,j}| \leq 1$.

On supp page 36, for the sentence "Since we know that the BAF for the diploid/tetraploid cluster s is approximately 0.5, we proportionally correct the BAF of every cluster with mirrored BAF ...", please provide very exact formula as well.

On supp page 42, in the definition of Problem 3, specify θ . I am aware that it has been discussed before this definition, but to make the definition mathematically sound it needs to be mentioned inside it that θ is a given constant. Do the same at the other places where needed, like at the end of the first paragraph on supp page 43.

I could not find instructions for installing this tool on Windows OS. Is this going to be provided in the future? If not, HATCHet would not be the only tool missing such specifications and I would be fine then with testing this on Linux, but please just clarify it.

Interestingly, the following is stated in the repository "Gurobi is a commercial ILP solver with two licensing options: (1) a single-host license where the license is tied to a single computer and (2) a network license for use in a compute cluster (using a license server in the cluster). Both options are freely and easily available for users in academia \url{here}."

This is the first time that I hear from someone that setting up free compute cluster license for Gurobi is so easy and straightforward. In the past years several of our collaborators tried to set Gurobi on their clusters with varying success, some of them giving up and for the others it took a long time and was everything but not easy (assuming no money paid). Maybe something has changed in the past few months or their institutions had some specific requirements. Anyway, I recommend toning this sentence down. On the other hand, from my personal experience setting up free Gurobi license on a private computer (e.g. Windows laptop) is quite easy for students and others eligible to apply for it.

Are related previous tools from this group, THetA and THetA2, now obsolete? If they are, can you please make sure that this is very clearly indicated in their repositories and that appropriate link to this new tool is provided. Adding this information to the introduction in README in HATCHet's repository would also be beneficial. In this crowded field it is very important that the user community can easily identify the best performing tool of this group and be spared of wasting time on running out of date tools. What about the tool accompanying the publication "Phylogenetic Copy-Number Factorization of Multiple Tumor Samples" mentioned above? If it is not obsolete, how does it compare to HATCHet and are there clear recommendations how to decide which one of them to use?

(Minor) In the caption of Figure 3: "HATCHet and Battenberg infers", should be "infer".

(Minor) In the caption of Figure S37, "... and a average error lower than Battenberg ..." should be "... and an average error lower than Battenberg ...".

(Minor) In the caption of Figure S11, correct the following sentence: "Methods based on a clone-specific model group CNAs into close and model the specific proportion of every clone."

(Minor) Supplementary Material, section A.1.: why is the total number of reads indexed by sample (it is denoted by R_p), whereas the total number of cells is not (it is denoted by E)? This is not necessarily wrong, but it seems as inconsistent and introduces unnecessary confusion. Also, adding the expected approximate value of L_1 , which is around 6 billion bases for the human genome, would help reader in better understanding of L_1, \dots, L_n (this is mentioned later "... with respect to the genome length L_1 of the normal cells, that is twice the reference length, i.e. $L_1 = 2L$.", but I recommend mentioning it as soon as L_i 's are introduced).

(Minor) Supplementary Methods, page 31, please revisit the sentence: "More specifically, we first define the read-depth ration (RDR) and we model the fractional copy numbers to show that their are directly proportional."

(Minor) There is no need to repeat definitions of several variables after Supp Equation (19).

(Minor) In "... as previously reported in the prostate publication", "prostate publication" sounds inappropriate.

(Minor) Supp page 43 "We design a ILP" -> "We design an ILP". Same on supp page 44.

(Minor) Is one of "the" and "our" redundant in "A detailed description of our the model selection procedure is in Supplementary Note B.4." ?

Reviewer responses for manuscript NCOMMS-20-03751-T

Simone Zaccaria¹ and Benjamin J. Raphael^{1,*}

¹Department of Computer Science, Princeton University, Princeton, NJ 08540

*Correspondence: braphael@princeton.edu

We thank the reviewers for their detailed and thoughtful comments. In response to their comments, we have added several new analyses with detailed descriptions. We have also corrected a few minor errors and typos in the manuscript. We highlight substantial changes to the manuscript in blue text. Below we provide a point-by-point response (blue text) to each of the reviewer comments (black text). All references to sections and figures refer to the revised version of the manuscript, including the revision of the main text and the revision of the Supplementary Note.

Reviewer 3

“We thank the authors for their detailed responses to our initial comments--the authors have clarified most of the concerns by adding more comparisons between known software tools (including Canopy, Battenberg and ReMixT), as well as using somatic SNVs and indels to support their findings. Many of the provided explanations are much clearer. In particular, the author responses helped highlight what HATCHET and MASCOTE do differently compared to other methods, so the novelty of the methods is more apparent. Table R1 is a great addition to clarify not only how HATCHet differs from existing methods but also how to conceptually call SCNAs from bulk WGS data (particularly for readers relatively new to the field).

The authors do concede that cloneHD has implemented joint-sample calling to detect CNAs, but they stress that HATCHet is one of the few methods that carries out joint-sample calling and to better effect than cloneHD (with advantages over other methods). Based on the revision and response, it appears that the chief innovations of HATCHet are (1) to determine which segments of the genome share the same copy number state on the basis of a clustering algorithm applied to only BAF and RDR (regardless of WGD) and (2) to determine whether WGD occurred until after estimating allele-specific fractional copy numbers and clonal fractions in order to better handle the uncertainty of calling CNAs in samples with possible WGDs -- which does represent a useful insight. Otherwise, the revised manuscript is much clearer, particularly the methods section which describes HATCHet's mathematical logic and design choices more plainly. The manuscript as

presented may be suitable for publication, but we have the following comments for the authors to consider.”

We thank the reviewer for the positive evaluation of our manuscript.

“Major comments

1. Figure 6 describes using HATCHet to explain somatic SNV allele frequencies after CNAs and WGD events. However, there could be a number of somatic SNVs that occurred after the CNAs. While the authors treated mutations as “explained” if only when their VAFs were within 95% CI of predicted VAFs from mutations occur before the CNAs, a large proportion of somatic SNVs could arise later than the CNAs and would have much lower VAFs highly deviated from their expectations. As a result, these SNVs could not serve as a good dataset to compare the performance of different methods. Is it possible to use read-level phasing information of allele-specific CNAs and the linked somatic variants to do a better evaluation? An analysis of heterozygous polymorphisms within regions affected by CNAs would provide a more convincing benchmark -- the true minor-allele fraction of hSNPs within genomic duplication should be inversely proportional to the estimated CN of the region. Additionally, any somatic SNVs phased to a nearby hSNP should show similar results as the hSNP in the proposed simulation.”

We agree with the reviewer that some of the somatic SNVs occurring after the CNAs at the same locus may have VAFs that are lower than expected and thus cannot be explained by CNAs. To provide a more fair comparison of the methods, we performed an additional analysis including two points that specifically address the mentioned issue. First, we excluded mutations that cannot be explained by the copy numbers and proportions inferred by *any* of the methods. Thus, we assume that if there is a combination of copy numbers and proportions (from either HATCHet, Battenberg, or Control-FREEC) that explain a mutation, then the mutation can be explained by CNAs. Second, we excluded low-frequency mutations ($VAF < 0.2$) which are the most-likely mutations affected by the mentioned issue. We added the corresponding description in Section 2.6:

“When counting the number of explained mutations, we excluded mutations that have low frequency ($VAF < 0.2$) as well as mutations that are not explained by the copy numbers and proportions inferred by any of the methods. These excluded mutations are more likely to have occurred after CNAs and to be present in smaller subpopulations of cells.”

While the number of unexplained mutations has been substantially reduced (the reduction across patients is of 39-60% in the prostate dataset and 23-47% in the pancreas dataset) from the previous version of Fig. 6 (as expected by the reviewer), we observed that HATCHet still explains a substantially higher number of mutations than Battenberg and Control-FREEC on both the prostate

and pancreas cancer datasets. We report the new results in the revised Fig. 6B-C (reproduced in Fig. R1B-C below).

Fig. R1: HATCHet infers copy-number states and proportions that better explain variant allele frequencies (VAFs) of somatic single-nucleotide mutations. (A) A genomic segment (cyan rectangle) harbors a somatic single-nucleotide mutation. Reads with alternate allele (red squares) and reference allele (grey squares) are used to estimate the VAF. (Top right) From T sequencing reads (gray rectangles) covering the mutation, a 95% confidence interval (CI, i.e. red area of posterior probability) on the VAF is obtained from a binomial model. (Bottom) Separately, copy-number states and proportions are inferred for this genomic segment. Given the numbers \tilde{c}_1, \tilde{c}_2 of mutated copies of the mutation in each of the two copy-number states, the VAF of the mutation is computed as the fraction of the mutated copies weighted by the proportions of the corresponding copy-number states. Assuming that an allele-specific position is mutated at most once during tumor progression (i.e. no-homoplasmy), all possible values of VAF are computed according to the possible values of \tilde{c}_1, \tilde{c}_2 . A mutation is explained if at least one value of VAF is within CI. (B) On the prostate dataset, HATCHet’s copy numbers (red) yield fewer unexplained mutations than Battenberg (blue) in all patients but A29, where the difference is small. (C) On the pancreas dataset, HATCHet’s copy numbers yield fewer unexplained mutations in all patients than Control-FREEC.

The reviewer’s suggestion to phase SNVs and germline heterozygous SNPs is a good one. Unfortunately, we found that extremely small fractions of the high-confidence mutations used in the SNV analysis are close enough to germline SNPs for phasing: <0.2% in the prostate cancer patients and <0.17% in the pancreas cancer patients. Phasing of this small number of SNVs would not substantially change the fraction of explained mutations for each method. We noted these considerations in a new paragraph of Discussion:

“In this work we showed that the copy numbers and clone proportions inferred by HATCHet across multiple samples allow a better explanation of somatic SNVs and indels in both the prostate and pancreas cancer patients. In particular, HATCHet yielded more reasonable estimates of CCFs of the SNVs in the prostate cancer patients with a critical impact on the analysis of tumor evolution. Further improvements integrating copy numbers and SNV analyses are an important future direction. For example, phasing somatic mutations to nearby germline SNPs might provide additional information to identify explained mutations, although in the present study, only a small fraction of the mutations (<0.2% in the prostate and <0.17% in pancreas cancer patients) are on the same sequencing read as a heterozygous germline SNP.”

Lastly, the reviewer’s suggestion to analyze the minor-allele fraction (namely the B-allele frequency (BAF)) of germline polymorphisms (i.e. SNPs) is also a good one. We already performed three analyses using BAFs to assess the copy numbers and proportions inferred by HATCHet. First, we have used the shift of BAF from BAF=0.5 – the expected value for allelic balanced regions – to show that the clonal CNAs inferred by HATCHet are more supported than the subclonal CNAs inferred by Battenberg in the prostate cancer patients (see x-axis in Fig. 3C and 3E as well as Supplementary Fig. S13). Second, we used the same shift in BAF to show that the subclonal CNAs inferred by HATCHet are more supported than the clonal CNAs inferred by Control-FREEC on the pancreas cancer patients (see x-axis in Fig. 4B-D and Fig. 5B). Third, we introduced a novel distance function (called clonality distance) that uses BAF values (together with RDR values) to provide further evidence to the subclonal CNAs and WGDs inferred by HATCHet (Supplementary Results E.6).

“Minor comments

1. In Figure 3E, why did HATCHet call the small “cloud” to the left of (1,1) as “clonal”? It looks very subclonal (and I’m guessing it corresponds to the dip near chr11). The cloud could represent a sequencing artefact, given how the (2,1) region nearby also appears to dip.”

The small cloud to the left of (1, 1) does correspond to the dip near chr11. We believe that this dip is due to sequencing artefacts, as there is no shift observed in BAF (x-axis). Both HATCHet and Battenberg infer that the small cloud belongs to the clonal cluster with allele-specific copy numbers of (1, 1).

“Additionally, it seems strange that Battenberg would deem the (2,1) cluster of bins as a subclonal region when it had no such problem in the B/C. Is Battenberg being thrown off by coverage biases?”

Battenberg deems the (2,1) cluster of bins as subclonal both in sample A10-C in Fig. 3C and in sample A10-A in Fig. 3E. However, Fig. 3C highlights (magenta) the regions that are identified as subclonal by *both* HATCHet and Battenberg, while Fig. 3E highlights (green) the regions that are identified as subclonal *only* by Battenberg.

To highlight the additional subclonal CNAs identified only by Battenberg in A10-C, we added a new Supplementary Fig. S14 in the manuscript (reproduced in Fig. R2 below) which highlights in green the regions in A10-C that are identified as subclonal only by Battenberg (panels (C) and (D)). To facilitate a visual comparison, we also highlight in magenta in panels (A) and (B) of the new Fig. S14 the regions in sample A10-C that are identified as subclonal CNAs by both HATCHet and Battenberg.

Fig. R2: Battenberg overestimates the presence of subclonal CNAs in the sample A10-C of prostate cancer patient A10. (A) In sample A10-C of patient A10, both HATCHet and Battenberg identify reliable subclonal CNAs that correspond to sample-subclonal clusters (magenta) with positions in the scaled BAF-RDR plot (each point corresponds to 50kb genomic bin) that are clearly in between the positions of sample-clonal clusters (black clusters with corresponding copy-number states). (B) The sample-subclonal clusters in (A) correspond to large genomic regions (magenta) with values of RDR (for 50kb genomic bins) that are clearly distinct from the RDR values of regions from sample-clonal clusters (black). (C) In the same sample A10-C, Battenberg identifies extensive clusters of genomic bins with subclonal CNAs (green). However, the clusters corresponding to these subclonal CNAs are not clearly distinguished in the scaled RDR-BAF plot (each point corresponds to 50kb genomic bins) from sample-clonal clusters (black). Thus, HATCHet only identifies clonal CNAs in this sample. (D) The sample-subclonal clusters in (C) correspond to large genomic regions (green) with values of RDR (for 50kb genomic bins) approximately equal to the RDR values of nearby regions from sample-clonal clusters (black).

“2. For figures 4 and 5, it would help to show which of the clusters (in panels B onward) were called in the published analysis and what were the estimated copy number states for the clusters that HATCHet did deem as subclonal or arising from a WGD. As presented, figures 4 and 5 are not clear enough to visually show what unique information about SCNAs would HATCHet provide, so some additional annotation would be helpful”

We have revised Fig. 4 and 5 in the manuscript (reproduced in Fig. R3 and R4 below) to include the copy numbers inferred in the published analysis and to clarify the unique information provided by HATCHet. Specifically, we made four major changes. First, we used additional annotations (arrows and red squares) to highlight the novel subclonal CNAs or WGD-related copy numbers revealed by HATCHet. Second, we swapped the axes of the scaled RDR-BAF plots: RDRs (or fractional copy numbers) are now on the y-axis. Although this is not directly related to the reviewer’s comment, we believe that this representation provides a better alignment with standard RDR plots as in Fig. 4E. Third, we indicated the total copy numbers inferred by Control-FREEC in the published analysis for the different clusters on the right side of the scaled BAF-RDR plots in Fig. 4B and Fig. 4E as well as in Fig. 5B. Lastly, we highlighted with black dashed lines the values of fractional copy numbers corresponding to clonal CNAs. This change should help the reader to easily distinguish the clonal clusters that lay on these black lines from the subclonal clusters that are located in between these lines.

Fig. R3: HATCHet identifies well-supported subclonal CNAs in metastatic pancreas cancer patients. (A) HATCHet identifies subclonal CNAs in 15 of 35 samples, while published analysis used Control-FREEC and excluded subclonal CNAs. **(B)** In the lymph node metastasis sample Pam01_NoM1, HATCHet infers two distinct tumor clones (ellipses in lower right of plot with corresponding proportions) and a tumor purity of 69%. Five sample-subclonal clusters (arrows) of 50kb genomic bins occupy intermediate positions between the other sample-clonal clusters (dashed black lines) in the scaled BAF-RDR plot, and thus have distinct copy-number states in the two clones, corresponding to subclonal CNAs. Control-FREEC copy numbers are shown on right y-axis labels. **(C)** In a second liver metastasis sample Pam01_LiM2 from the same patient, HATCHet infers two

distinct tumor clones, one (red) shared with the lymph node sample Pam01_NoM1. A large sample-subclonal cluster (brown, starred) occupies an intermediate position in the scaled BAF-RDR plot and has distinct copy-number states in the two clones. In contrast, the five sample-subclonal clusters in Pam01_NoM1 (arrows) clearly overlap the sample-clonal clusters in this sample and thus correspond to clonal CNAs (dashed black lines). **(D)** In the liver metastasis sample Pam01_LiM1, HATCHet identifies a single tumor clone (white) that is shared with the lymph node metastasis sample Pam01_NoM1 in (B). The five sample-subclonal clusters in Pam01_NoM1 (arrows) correspond to clonal CNAs in sample Pam01_LiM1 but have different copy-number states than those in (C). The inferred low tumor purity (28%) of this sample results in a partial overlap of clusters that are clearly distinguished in higher purity samples in (B) and (C). **(E)** The five sample-subclonal clusters in Pam01_NoM1 (arrows) correspond to large genomic regions with values of RDR that are clearly distinct from the other sample-clonal clusters (dashed black lines). Genomic regions that are part of small clusters or have out-of-scale values are reported in gray. Ranges of fractional copy numbers corresponding to the total copy numbers inferred by Control-FREEC in the previously published analysis are shown on right y-axis labels.

Fig. R4: HATCHet identifies WGDs in three of four pancreas cancer patients. (A) HATCHet predicts a WGD in all 31 samples from 3 patients (Pam01, Pam02, and Pam03). In contrast, published analysis used Control-FREEC and excluded WGDs. **(B)** In four samples of patient Pam02, HATCHet predicts a WGD and infers two tumor clones (ellipses in upper right of plot with corresponding proportions) with 7 large tumor-clonal clusters (arrows with corresponding copy-number states). These clusters preserve their relative positions in the scaled BAF-RDR plot (each point corresponds to 50kb genomic bin) across samples and their fractional copy numbers correspond to sample-clonal clusters in each sample (dashed black lines), supporting the inference of a tumor-clonal CNA (i.e. unique copy-number state across samples) for each of these clusters. Note that without a WGD three clusters (red dashed squares) would correspond to subclonal CNAs in all samples. Two additional clusters (peach and olive, starred) are tumor-subclonal as they change their relative position across samples (Pam02_PT18 and Pam02_LiM4 vs. Pam02_LiM3 and Pam02_LiM5), supporting the inference of two distinct tumor clones in this patient. The total copy numbers inferred by Control-FREEC in published analysis are shown on right y-axis labels in the first scaled BAF-RDR plot.

Furthermore, we added a new Supplementary Figure S17 in the manuscript (reproduced in Fig. R5 below) to show the published copy numbers inferred by Control-FREEC for the same clusters and cancer samples depicted in Fig. 4 and 5.

(A) Pancreas patient Pam01

(B) Pancreas patient Pam02

Fig. R5: Published copy numbers derived from Control-FREEC for pancreas cancer patients Pam01 and Pam02 are inconsistent across samples and miss subclonal CNAs and WGDs. (A) RDRs and BAFs of 50kb genomic bins in three samples from the pancreas cancer patient Pam01 are colored according to the published total copy numbers inferred by Control-FREEC. (B) RDRs and BAFs of 50kb genomic bins in four samples from the pancreas cancer patient Pam02 are colored according to the published total copy numbers inferred by Control-FREEC.

Reviewer 4

“HATCHet, a method for studying tumor clonal architecture from multi-sample bulk sequencing data and using copy number aberrations as identifiers of individual tumor clones, is introduced. Similarly to many existing methods, in particular those designed for SNVs, HATCHet uses all samples simultaneously and assuming that each bulk tumor sample is a mixture of distinct clones (I will refer to each population of cells with unique copy number profile as clone). The set of clones typically consists of a few clones (in the simulated data up to 3 tumor clones), with one of them representing population of healthy cells, whereas the others are tumor clones characterized (defined) with unknown copy number profiles. Clones are assumed to be shared among bulk samples, with varying proportions, which is a standard assumption well motivated by tree of tumor evolution that is common to all samples. It is allowed that some clones are absent from a given sample, but if a clone is present then at least u_{\min} fraction of cells in the sample are required to

belong to this clone, where u_{\min} is a parameter. In addition to inferring copy number profiles of clones, method also infers the number of clones and their prevalence in tumor samples. Two distinct cases are considered when inferring the unknowns:

1. case where it is assumed that no whole genome duplication (WGD) occurred
2. case where it is assumed that WGD occurred

Using selection criterion it is decided which of the two is more likely and the number of clones is selected as well. HATCHet is not limited to working with the total copy number, but distinguishes between minor and major alleles by the use of standard B-allele frequencies computed at the sites of heterozygous SNPs.

Looking into the method, genome is first segmented into small bins and read-depth ratio (RDR) and B-allele frequency (BAF) values are computed for each bin in each sample. These values are then combined in a specific order in vector of dimension $2k$, where k is the number of bulk samples (each sample gives one RDR and one BAF value) and clustered through BPNY (reference provided in the Supp Material). During this clustering, bins are clustered into m clusters, with number of clusters inferred by the clustering algorithm. Each cluster consists of a set of genomic bins sharing similar BAF and RDR values across all samples. Note that the inferred clusters do not necessarily contain only block of adjacent bins but, for example, bins from different chromosomes can be clustered together as soon as their BAF and RDR values are similar across all samples. The algorithm then mostly works with the obtained clusters instead of bins. Assuming correct clustering, RDR and BAF estimates of clusters are expected to be more stable than these of individual bins. In addition, clustering also leads to significant dimensionality reduction as the number of clusters is usually small in comparison to the number of bins.

After that, fractional copy numbers are estimated. This step requires sequencing data of matching normal (healthy) tissue and assumes that a good fraction of the genome is diploid and not affected by copy number aberrations if no WGD is present (similar assumption is made in case where WGD is present). The fractional copy number values are then used as the input to another important part of the algorithm where allele-specific copy numbers of all segments are inferred together with clonal frequencies. This inference is done using, from the algorithmic point of view, minor modifications of previously published ideas and solutions based on coordinate-descent algorithm for some of the problems presented in "Phylogenetic Copy-Number Factorization of Multiple Tumor Samples" (by the same authors, published in Journal of Computational Biology). Note that this was later also used in "Deconvolution and phylogeny inference of structural variations in tumor genomic samples" (Eaton et al., Bioinformatics, 2018). In addition to the coordinate-descent algorithm, the optimization problem, which is not solvable by most (if not all) of the available solvers in the introduced form, is transformed to an instance of classic ILP using standard linearization techniques."

We thank the reviewer for the thorough summary of our algorithm.

"Assume, for simplicity of notation, that in this paragraph of the review $u = u_{ip}$ and $x = x_{ip}$. For $u_{\min} > 0$ (which is a case, otherwise having u_{\min} is meaningless), combination of constraints (46)

and (47) given in the Supplementary Material would force that all x's are set to 1. Namely, constraint (46) can be rewritten as

$$u \geq u_{\min} + (1-x)$$

so setting variable x to 0 would imply that $u \geq 1 + u_{\min} > 1$, which is impossible due to the defined range of variables u (as well as due to the constraint (45)). Alternatively, x=0 would enforce that u=0 through (47) and then (46) will not be fulfilled. Therefore x would never be set to 0 enforcing that each clone is present in each sample that is obviously unrealistic requirement.

The correct system is:

$$u + 1 - x \geq u_{\min}$$

$$u \leq x$$

Alternatively, one can impose the following constraints:

$$u \geq u_{\min} * x$$

$$u \leq x$$

I hope that this is just one of numerous typos in the manuscript (see below). It is definitely very bad place to have typos as this would most likely waste hours of time of the potential user interested in re-implementing this part of the tool (such re-implementation might be needed if, for example, user does not have Gurobi license and wants to re-implement this in other optimization software, like CPLEX or some freely available ILP solver). If this is not a typo, the authors might need to re-run the experiments performed.”

The error was indeed a typo in the manuscript and we thank the reviewer for catching it. The correct equations are those stated by the reviewer, and we corrected the equations in the manuscript accordingly.

“While considering the possibility of existence of whole genome duplications (WGDs), three main assumptions about WGDs are made: (i) WGDs occur at most once (ii) if WGD occurs, then it affects all tumor clones (iii) if WGD is present, there also exist some other clonal copy number aberrations. While each of the assumptions is not necessarily always true in practice, I find the first two of them quite fair. For example, there is indeed evidence in the literature that WGDs are early events in carcinogenesis. The third assumption is questionable as single WGD combined with SNVs, indels, fusions and/or translocations can suffice for cancer formation. Although future studies, primarily those involving single-cell data, will provide better insight on this, I am of the opinion that it is more likely that this assumption is true than not in most cases, but I recommend adding very brief discussion on the potential limitation of the method due to this assumption.”

We added a corresponding comment in Discussion:

“Third, HATCHet’s modeling of WGD could be further generalized. While recent pan-cancer studies^{5-8, 12} show that the current assumptions used in HATCHet (namely that a WGD occurs at most once as a clonal event and that additional clonal CNAs also occur) are reasonable for most tumors,

HATCHet's model could be extended to allow for multiple WGD (e.g. hexaploid or higher ploidy), subclonal WGD, or a WGD occurring without any other clonal CNAs."

"Why the inference of similar fractions of the genome affected by CNAs implies that subclonal CNAs inferred by Battenberg are clonal (see sentence "Second, both methods inferred similar fractions of the genome affected by CNAs on these 20 samples (Fig. S34A), suggesting that the subclonal CNAs inferred by Battenberg are clonal instead")?"

We used the difference between the fractions of the genome with CNAs to quantify the difference between the proportions of the genome that contain subclonal CNAs, as inferred by Battenberg and by HATCHet. We observed that many of the subclonal CNAs identified by Battenberg are inferred to be clonal by HATCHet, especially in 20 prostate cancer samples where HATCHet only infers clonal CNAs (Fig. 38). Since Battenberg does not fit the observed RDRs and BAFs better than HATCHet (Fig. S41), the additional subclonal CNAs inferred by Battenberg can be equally explained as clonal CNAs. To clarify this point, we have revised the corresponding paragraph in the manuscript:

"While it is possible that Battenberg has higher sensitivity in detecting subclonal CNAs than HATCHet, the extensive subclonal CNAs reported by Battenberg in all samples is concerning. This is because the inference of subclonal CNAs will always produce a better fit to the observed RDRs and BAFs, but with a cost of increasing the number of parameters required to describe the copy-number states (model complexity). Battenberg models the clonal composition of each segment independently (Fig. S15), and thus has 6X more parameters than HATCHet on this dataset (Fig. S42). To avoid overfitting, it is important to evaluate the trade-off between model fit and model complexity. Battenberg does not include a model-selection criterion to evaluate this trade-off, and it consequently infers a high fraction of subclonal CNAs in every sample (Fig. S38) without fitting the observed RDRs and BAFs better than HATCHet (Fig. S41). In contrast, HATCHet uses a model-selection criterion to identify the number of clones; consequently in 20/49 samples HATCHet infers that all the subclonal CNAs identified by Battenberg are instead clonal (Fig. S38). Since HATCHet fits the observed RDRs and BAFs as well as Battenberg (Fig. S41) but without subclonal CNAs, the extensive subclonal CNAs reported by Battenberg in these samples are equally well-explained as clonal CNAs."

"The next sentence, "Finally, we found that ReMixT's inference of subclonal CNAs from the same dataset was more similar to HATCHet than Battenberg". What is the main message that one should take from this sentence?"

According to the benchmark on simulated data, both HATCHet and ReMixT outperformed Battenberg (Section 2.2). Therefore, the fact that ReMixT infers results more similar to HATCHet than Battenberg suggests that the results of HATCHet and ReMixT are more accurate. Moreover, we emphasize that the comparison between HATCHet and ReMixT on the prostate cancer dataset has

been introduced in response to the previous requests of Reviewer 2. We have correspondingly revised the sentence as follows:

“Finally, we found that ReMixT’s inference of subclonal CNAs from the same dataset was more similar to HATCHet than Battenberg (Fig. S40 and Supplementary Note E.2). Since both HATCHet and ReMixT outperformed Battenberg on the simulated data, the similarity between HATCHet and ReMixT on this dataset suggests that Battenberg’s results are less accurate.”

“Why is comparison of results of HATCHet and Control-FREEC on pancreatic cancer dataset relevant? Can available alternative methods give the same insight as HATCHet into clonal composition of these tumors?”

The main point in the analysis of the pancreas dataset is *not* to directly compare HATCHet with Control-FREEC but rather to investigate whether subclonal CNAs and WGDs are present in the pancreas cancer samples. Since Makohon et al. (2017) used Control-FREEC and excluded the possibility of subclonal CNAs and WGDs, we wanted to evaluate whether HATCHet’s joint analysis of multiple samples and rigorous model selection of WGD would identify either of these features in the data. This is particularly interesting because WGD have been previously reported as common in pancreatic tumors [TCGA, *Cancer Cell* 2017]. We clarified this motivation in the first paragraph of Section 2.3:

“While both datasets contain multiple tumor samples from individual patients, the previously published analyses inferred CNAs in each sample independently. Moreover, these studies reached opposite conclusions regarding the landscape of CNAs in these tumors. Gundem et al.¹¹ report subclonal CNAs in all primary and metastatic prostate samples. In contrast, Makohon-Moore et al.³⁰ report no subclonal CNAs in the primary and metastatic pancreatic samples. An important question is whether this difference is due to cancer-type specific or patient-specific differences in CNA evolution of these tumors, or a consequence of differences in the bioinformatic analyses. We investigated whether the HATCHet’s analysis would confirm or refute the discordance between the copy-number landscapes reported in these studies.”

We also clarified the motivation by adding the following paragraph in Section 2.4:

“We next examined the prediction of whole-genome duplications (WGDs) on the prostate and pancreas cancer datasets. The previously published analyses of these datasets reached opposite conclusions regarding the landscape of WGDs in these tumors. Gundem et al.¹¹ report WGDs in 12 samples of 4 prostate cancer patients (A12, A29, A31, and A32). In contrast, Makohon-Moore et al.³⁰ did not evaluate the presence of WGDs in the pancreas cancer samples, despite reports of high

prevalence of WGD in pancreas cancer⁴⁵. We investigated whether HATCHet analysis would confirm or refute the different prevalence of WGD reported in the previous studies.”

“Looking into Figure S1, how does blue subclone differ from the healthy cells? I see blue profiles twice, each time it is (1,1). Furthermore, why some of the clusters have black and blue profiles associated with them, whereas the others have only black? Overall, I do not find explanation (caption) of this figure clear enough.”

We previously labelled the clusters corresponding to clonal CNAs by only using a single pair of copy numbers (the one from the black/white clone), and we specified the copy numbers of the blue clone only for subclonal CNAs. However, we understand the confusion of the previous choice and we revised Fig. S1 in the manuscript (reproduced in Fig. R6 below) to include the copy numbers of both clones for every cluster. Moreover, we fully revised the caption of Fig. S1 to improve clarity.

Fig. R6: Interpretation of many clusters as subclonal CNAs vs. WGD. RDRs and BAFs for five clusters of genomic bins (colors) have two alternate explanations. (Bottom left) The first explanation has two distinct tumor clones (white and blue) with nearly the same clone proportions (48% and 47%) and the normal clone in low proportion (5%). In this explanation, three clusters (green, dark orange, and dark blue) correspond to clonal CNAs with the same indicated copy-number state in both clones and the two remaining clusters (light orange and light blue) correspond to subclonal CNAs with different copy-number states as indicated in each clone. (Bottom right) The second explanation has a single tumor clone (white) containing a WGD with the tumor clone in high

proportion (90%) and the normal clone in low proportion (10%). In this explanation, all five clusters correspond to clonal CNAs.

“The end of Section 2.4: Why is having one of WGD and subclonal CNAs, but not both of these, a good indication of the quality of solution? Can't WGD introduce higher genomic instability leading to the subsequent subclonal CNAs?”

We agree with the reviewer that subclonal CNAs and WGDs can be both present, and HATCHet makes this inference in some cases; see for example the HATCHet's results on sample A29-A of patient A29 (Fig. S19). However, while this is indeed a possibility, the inference of solutions with both subclonal CNAs and WGDs requires care. Inferring both subclonal CNAs and WGDs generally produces a better fit to the data than inferring only subclonal CNAs or only WGDs. This is because both subclonal CNAs and WGDs increase the total number of copy-number states available to explain distinct clusters (see response above). Following the parsimony principle (or most reasonable model selection procedures) the simpler explanation (with only subclonal CNAs or only WGD) is preferred.

We clarified the explanation of the paragraph mentioned by the reviewer in Section 2.4:

“The two discordant samples, A12-C and A29-C, are single samples from patients A12 and A29, respectively (Fig. S18A,B). Battenberg predicted a WGD only in A12-C and no WGD in the other samples from this patient. Conversely, Battenberg predicted no WGD in A29-C but a WGD in the other sample from this patient. However, the divergent predictions of WGD in only one sample of these patients is not well-supported by the data. In particular, the observation of a large number of distinct clusters of genomic bins (i.e. distinct copy-number states) in a sample has two reasonable explanations: subclonal CNAs or a WGD (Fig. S1). Since Battenberg analyzes each sample independently it may choose a different explanation (subclonal CNAs vs. WGD) for the large number of clusters observed in each sample from the same patient. In some cases like sample A12-C, Battenberg predicts both subclonal CNAs and WGD (Fig. S18C and Fig. S19A). Since both subclonal CNAs and WGDs increase the total number of copy-number states available to explain distinct clusters, they will generally provide a better fit to the data. However, there is a danger of overfitting since both WGD and subclonal CNAs increase the number of parameters in the model. In contrast, HATCHet jointly analyzes multiple samples and predicts the absence/presence of a WGD consistently across all samples from the same patient (Fig. S18A,B): no WGD in all samples of patient A12 and a WGD in all samples of patient A29. Moreover, HATCHet integrates the choice of WGD into the model selection procedure, providing a simpler explanation of the data (with only subclonal CNAs or only WGD) with an equally good fit to the observed RDRs and BAFs (Fig. S18D, Fig. S19B, and Fig. S60A).”

“Overall, while the results on real sequencing data to some extent demonstrate that considering multiple samples simultaneously might provide some advantage to clonal architecture

reconstruction method, many of the results are anecdotal, debatable and it is not very convincing why solutions reported by HATCHet are a good explanation of the observed read counts. Instead of comparing HATCHet's results with the results of available alternatives on datasets for which even no proxy of ground truth is available, have the authors considered analyzing copy number profiles of tumors for which copy number profiling at single-cell level was performed? A lot of work on this was done for example by Nicholas Navin, one of the pioneers of single-cell sequencing, and his group at MD Anderson Cancer Center. There are also many studies from the other groups and growing number of datasets with available single-cell and matching bulk sequencing data.”

We performed a new analysis comparing HATCHet’s results on bulk sequencing data with DOP-PCR derived single-cell copy-number profiles of 8 breast cancer patients from two publications from Nicholas Navin’s group. Specifically, we used HATCHet to analyze whole-exome sequencing data from 12 bulk tumor samples of 4 breast cancer patients (P6, P9, P14, and P11) from Kim et al. (2018) and whole-exome sequencing data from 9 bulk tumor samples of 4 breast cancer patients (P4, P5, P6, and P10) from Casasent et al. (2018). We observed a reasonable consistency between the single-cell copy-number profiles and the copy numbers derived by HATCHet from the bulk whole-exome sequencing data (Fig. S9 and Fig. S10 in the revised manuscript, reproduced in Fig. R7 and Fig. R8 below). Despite the consistency between the results, we note that this comparison has some limitations. First, identification of CNAs from whole-exome sequencing data is much more challenging than from whole-genome sequencing data since whole-exome sequencing targets <2% of the genome. Second, the clonal composition of the bulk samples and the single cell samples may be different. Lastly, the published DOP-PCR copy-number profiles are particularly noisy as DOP-PCR sequencing has very low coverage per cell (<0.3X). Although the reviewer is correct that increasing amounts of single-cell sequencing is being performed, many of these datasets are limited in their utility as a validation for HATCHet. We are not aware of any publicly-available dataset that contains high-coverage whole-genome sequencing from multiple bulk-tumor samples as well as single-cell DNA sequencing data from the same samples with sufficiently high and uniform sequencing coverage per cell.

We added the new analysis in Section 2.2 and we describe its details in the new section in Supplementary Note E.7:

“Finally, we further assessed the performance of HATCHet by comparing copy-number profiles derived by HATCHet on bulk-tumor sequencing data with copy-number profiles from DOP-PCR single-cell DNA sequencing data from the same tumors. Specifically, we used HATCHet to analyze whole-exome sequencing data of 21 bulk-tumor samples from 8 breast cancer patients: 12 bulk-tumor samples from 4 breast cancer patients (P6, P9, P14, and P11) from Kim et al. (2018) and 9 bulk-tumor samples from 4 breast cancer patients (P4, P5, P6, and P10) from Casasent et al. (2018). We compared the copy numbers inferred by HATCHet jointly across the 2-3 bulk-tumor

samples from each patient with the copy-number profiles inferred from the DOP-PCR single-cell sequencing data from the same patient.

We observed a reasonable consistency between HATCHet's results and the single-cell copy-number profiles (Fig. S9 and Fig. S10). Specifically, HATCHet correctly identified ~93% of the clonal CNAs reported in the single-cell copy-number profiles across all 8 patients. In 5/8 patients, HATCHet identifies a single tumor clone. In some of these patients, more than one distinct single-cell copy-number profile was reported, but most of these additional copy-number profiles were associated with a small fraction of cells (<7%). Such low prevalence profiles are difficult to detect in bulk tumor samples, and may be present at different frequencies in the bulk samples, since each bulk sample and the single cells are distinct collections of cells from the same tumor. In the other 3/8 patients, HATCHet identified multiple tumor clones. In particular, HATCHet identified subclonal CNAs in ~76% of the genomic regions where the single-cell copy-number profiles exhibit different copy numbers across cells. In most of these regions, HATCHet correctly identified the copy numbers of the most prevalent tumor clone. While HATCHet is unable to identify all subclonal CNAs found in the single-cell profiles, there are notable limitations in this comparison. First, identification of CNAs from whole-exome sequencing data is much more challenging than from whole-genome sequencing data since whole-exome sequencing targets <2% of the genome. Second, the clonal composition of the bulk samples and the single cell samples may be different. Third, the published DOP-PCR copy-number profiles are particularly noisy as DOP-PCR sequencing has very low coverage per cell (<0.3X).

Finally, we emphasize that our simulator MASCoTE was developed to generate realistic simulated multi-sample DNA sequencing data with full-known ground truth (Section 2.2 and Section 4.2). While these simulations may not reflect the full spectrum of variability in real data, the ground truth of each simulated dataset is fully known and the simulated datasets are not characterized by the high levels of noise and the sample biases found in the single-cell sequencing datasets. Thus, MASCoTE provides a reasonable approach to assess the performance of copy-number deconvolution methods without uncertainty on the ground truth.

Fig. R7: HATCHet's copy numbers are consistent with published single-cell copy-number profiles of 4 breast cancer patients. (Top of each panel) Single-cell copy-number profiles for each clone identified in DOP-PCR single-cell sequencing data from 4 breast cancer patients in Kim et al. (2018). On the right are the proportion of cells assigned to each clone. (Bottom of each panel) Total copy numbers inferred by HATCHet using whole-exome sequencing data from 2-3 bulk tumor samples (OP, 0, and 2) from each patient. On the right are the clone proportions inferred by HATCHet.

Fig. R8: **HATCHet's copy numbers are consistent with published single-cell copy-number profiles of 4 breast cancer patients.** (Top of each panel) Single-cell copy-number profiles for each clone identified in DOP-PCR single-cell sequencing data from 4 breast cancer patients in Casasent et al. (2018). On the right are the proportion of cells assigned to each clone. (Bottom of each panel) Total copy numbers inferred by HATCHet using whole-exome sequencing data from 2 bulk tumor samples (DCIS and INV) from each patient. On the right are the clone proportions inferred by HATCHet.

“I also have many other comments and suggestions related to the manuscript and the repository and they are listed below.

Isn't the variable forming the left hand side of Supp Equation (99) already explicitly defined in the few lines above as a variant allele frequency of SNV e in sample p ? I think that the current presentation is very confusing. On one side of Supp Equation (99) we have a very clearly defined VAF value, which is given as a function of variant and total read counts that are directly observed. Then, this value is expressed as a function of variables f , c and u through Supp Equations (99) and (100), where u 's are unknowns and f and c are defined at the level of cluster of segments (and their values depend on set of reads spanning many different genomic coordinates).”

This was a typo and Supplementary Eq. (99) was meant to model the predicted VAF $\bar{\psi}$, while the observed VAF ψ is indeed defined as the fraction of variant reads (see Fig. R1A above). We clarified these definitions in the text and corrected the corresponding typos (note that previous Eq. (99) is now Eq. (98)):

“We observe the variant-allele frequency (VAF) $\psi_{e,p}$ of every mutation e from every sample p as the fraction of reads in p harboring e at the corresponding locus (Fig. S52 and Fig. S53, respectively). When the mutation e is located in a genomic region belonging to a cluster s , we model the predicted VAF $\bar{\psi}_{e,p}$ of e in p similarly to the BAF $\beta_{s,p}$ in Section A.2 as the following

$$\bar{\Psi}_{e,p} = \frac{f_{s,p,e}}{f_{s,p}} \quad (98)$$

where $f_{s,p,e}$ is the mutated fractional copy number correspondingly equal to

$$f_{s,p,e} = \sum_{2 \leq i \leq n} c_{s,i,e} u_{i,p} \quad (99)$$

where $c_{s,i,e}$ is the related mutated total copy number for every clone i , i.e. the number of copies of s that harbor the mutation e over the total copy number $c_{s,i}$.”

“In Supp Equation (101), why $f_{\{s,p\}}$ is used in denominator? In other words, why elements of matrix F are used instead of elements of the product $(A+B)U$ of inferred matrices?”

We agree with the reviewer and we corrected the equation accordingly.

“On page 16, in sentence “We also define the corresponding distance for F^B , B , and b .”, what is b ? Should this be U ?”

Yes, we corrected it accordingly.

“How do the authors justify use of minimum subclonal prevalence as high as 15% for several patients analyzed in real data section? How should a user set this important parameter?”

The minimum clone proportion as high as 15% has been only applied to 2 prostate cancer patients and 2 pancreas cancer patients where there were 1-2 sequenced samples with extremely noisy sequencing data. The approach that we adopted was to increase the value of the minimum clone proportion u_{min} when the inferred solution was composed of tumor clones with proportions exactly equal to u_{min} , which may generally indicate overfitting. Further details of the experimental procedure adopted to choose these values is described in Supplementary Section E.1 (“Experimental setup”) and in the new Supplementary Table S3.

“In Supp Equation (1) denominator seems to be incorrect. Similarly, I think that the left hand side in Supp Equation (7) has a typo ($b \rightarrow s$).”

We corrected both equations accordingly.

“What is the purpose of adding Supp Equation (11)?”

We agree that explicitly defining the A-specific fractional copy number is not necessarily needed and we removed the old Supplementary Eq. (11).

“In Supp Equation (17), undefined variables are used in the nominator ($\hat{u}_{i,p}$), in LaTeX notation.”

We corrected the typo $\hat{u}_{i,p}$ into $u_{i,p}$ accordingly.

“In Figure S11, part A, right part (Copy-number profiles), for segment s_5 , why is it composed of 70% of total copy number 3 and 50% of total copy number 4 when 70% of (2, 1) and 20% of (3, 1) are shown in the left.”

The proportions (70%, 50%) had a typo and we corrected those into (70%, 20%) accordingly.

“In the introduction in Supp Section B.2., there is an incomplete part of the sentence: "i.e. a cluster s is tumor-clonal if $(a_{\{s,2\}}, b_{\{s,2\}}), (a_{\{s,3\}}, b_{\{s,3\}}), \dots, (a_{\{s,n\}}, b_{\{s,n\}})$.””

We completed the sentence as follows:

“i.e. a cluster s is tumor-clonal if $|\{(a_{s,2}, b_{s,2}), (a_{s,3}, b_{s,3}), \dots, (a_{s,n}, b_{s,n})\}| = 1$.”

“In the sentence introducing Supp Equation (24), γ_p , which does not appear in the system, is mentioned. Please remind a reader of Supp Equation (8) and dependency of $f_{\{s,p\}}$, γ_p and $r_{\{s,p\}}$. All in all, I recommend expanding $f_{\{s,p\}}$ based on Equation (8) and discussing what are known (observed) and unknown variables. When it comes to solving the system, it is obviously very trivial and straightforward task and full details of solving it can be omitted (as is already the case). Furthermore, the authors talk about unknown μ_p (before Supp Equation (24)) and then after Supp Equation (24) they provide formula for τ_p , yet another mistake.”

We applied all the suggested changes: we reminded the reader of Supplementary Eq. (8), we expanded $f_{s,p}$ in the system based on Supplementary Eq. (8), we clearly stated which values are either known or unknown, and we corrected the typo τ_p into μ_p .

“In the sentence starting on 5-th line on supp page 36, sentence should end with bin t , not bin b ?”

We corrected b into t accordingly.

“At the very top of supp page 42, $|a_{s,i} - a_{s,j}| \leq 1$ appears twice. The second one should be $|b_{s,i} - b_{s,j}| \leq 1$.”

We corrected it accordingly.

“On supp page 36, for the sentence "Since we know that the BAF for the diploid/tetraploid cluster s is approximately 0.5, we proportionally correct the BAF of every cluster with mirrored BAF ...", please provide very exact formula as well.”

We revised the sentence to introduce the exact formula as well:

“Since we know that $\beta_{s,p} = 0.5$ for the diploid/tetraploid cluster s , we proportionally correct the BAF $\hat{\beta}_{z,p}$ of every cluster z with mirrored BAF approximately equal to 0.5 by a factor of $\frac{0.5}{\hat{\beta}_{s,p}}$, i.e.

$\beta_{z,p} = \hat{\beta}_{z,p}^{0.5}$, where $\hat{\beta}_{s,p}$, $\hat{\beta}_{z,p}$ are the average mirrored BAFs for the bins within s and z , respectively.”

“On supp page 42, in the definition of Problem 3, specify θ . I am aware that it has been discussed before this definition, but to make the definition mathematically sound it needs to be mentioned inside it that θ is a given constant. Do the same at the other places where needed, like at the end of the first paragraph on supp page 43.”

We added θ in all the problem definitions in both the main text and in supplementary material, and in any other relevant place. For example, we have the following for Problem 3:

“Problem 3 (Distance-based Constrained Allele-specific Copy-number Factorization (D-CACF) problem). Given the allele-specific fractional copy numbers F^A and F^B , a number n of clones, a maximum total copy number c_{max} , a minimum clone proportion u_{min} , and a constant value $\theta \in \{1, 2\}$, find allele-specific copy numbers $A = [a_{s,i}]$, $B = [b_{s,i}]$, and clone proportions $U = [u_{i,p}]$ such that: the distance $D = \|F^A - AU\| + \|F^B - BU\|$ is minimum; $a_{s,i} + b_{s,i} \leq c_{max}$ for every cluster s and clone i ; either $u_{i,p} \geq u_{min}$ or $u_{i,p} = 0$ for every clone i and sample p ; for every cluster s , either $a_{s,i} \geq \theta$ or $a_{s,i} \leq \theta$ for all clones i ; for every cluster s , either $b_{s,i} \geq \theta$ or $b_{s,i} \leq \theta$ for all clones i .”

“I could not find instructions for installing this tool on Windows OS. Is this going to be provided in the future? If not, HATCHet would not be the only tool missing such specifications and I would be fine then with testing this on Linux, but please just clarify it.”

HATCHet can be potentially run on Windows OS using an environment that supports C++ libraries (e.g. MSVC) and python libraries (e.g. using conda). However, at the present time, HATCHet has not been yet tested on Windows OS. We added such information in the “Current issues” section of the HATCHet’s repository and we will update it once the tests are performed.

“Interestingly, the following is stated in the repository “Gurobi is a commercial ILP solver with two licensing options: (1) a single-host license where the license is tied to a single computer and (2) a network license for use in a compute cluster (using a license server in the cluster). Both options are freely and easily available for users in academia \url{here}.” This is the first time that I hear from someone that setting up free compute cluster license for Gurobi is so easy and straightforward. In the past years several of our collaborators tried to set Gurobi on their clusters with varying success, some of them giving up and for the others it took a long time and was everything but not easy (assuming no money paid). Maybe something has changed in the past few months or their institutions had some specific requirements. Anyway, I recommend toning this sentence down. On

the other hand, from my personal experience setting up free Gurobi license on a private computer (e.g. Windows laptop) is quite easy for students and others eligible to apply for it.”

We applied the suggested change to the repository.

“Are related previous tools from this group, THetA and THetA2, now obsolete? If they are, can you please make sure that this is very clearly indicated in their repositories and that appropriate link to this new tool is provided. Adding this information to the introduction in README in HATCHet’s repository would also be beneficial. In this crowded field it is very important that the user community can easily identify the best performing tool of this group and be spared of wasting time on running out of date tools. What about the tool accompanying the publication “Phylogenetic Copy-Number Factorization of Multiple Tumor Samples” mentioned above? If it is not obsolete, how does it compare to HATCHet and are there clear recommendations how to decide which one of them to use?”

We believe that HATCHet supersedes some older tools from the same research group for calling CNAs from bulk-tumor samples, including THetA and THetA2, and we will add a corresponding statement to their repositories after the final release of HATCHet. In contrast, HATCHet does not supersede the CNT-ND method that we previously presented in “*Phylogenetic Copy-Number Factorization of Multiple Tumor Samples*” [Zaccaria et al., *JCB*, 2018] since the main goal of CNT-MD is to reconstruct the phylogenetic tree of the different tumor clones, differently than HATCHet.

“(Minor) In the caption of Figure 3: “HATCHet and Battenberg infers”, should be “infer”.”

We corrected the typo accordingly.

“(Minor) In the caption of Figure S37, “... and a average error lower than Battenberg ...” should be “... and an average error lower than Battenberg ...”.”

We corrected the typo accordingly.

“(Minor) In the caption of Figure S11, correct the following sentence: “Methods based on a clone-specific model group CNAs into close and model the specific proportion of every clone.””

We fully revised the caption of Fig. S11 (corresponding to Fig. S15 in the revised manuscript) and corrected the previous typos.

“(Minor) Supplementary Material, section A.1.: why is the total number of reads indexed by sample (it is denoted by R_p), whereas the total number of cells is not (it is denoted by E)? This is not

necessarily wrong, but it seems as inconsistent and introduces unnecessary confusion. Also, adding the expected approximate value of L_1 , which is around 6 billion bases for the human genome, would help reader in better understanding of L_1, \dots, L_n (this is mentioned later "... with respect to the genome length L_1 of the normal cells, that is twice the reference length, i.e. $L_1 = 2L$ ", but I recommend mentioning it as soon as L_i 's are introduced)."

We applied both the suggested changes.

"(Minor) Supplementary Methods, page 31, please revisit the sentence: "More specifically, we first define the read-depth ration (RDR) and we model the fractional copy numbers to show that their are directly proportional.""

We revised the sentence as follows:

"More specifically, we first define the read-depth ration (RDR) and we show that RDR is directly proportional to the corresponding fractional copy number."

"(Minor) There is no need to repeat definitions of several variables after Supp Equation (19)."

We removed the repeated definitions.

"(Minor) In "... as previously reported in the prostate publication", "prostate publication" sounds inappropriate."

We changed the sentence into:

"... as reported in the published analysis of the prostate cancer patients¹¹."

"(Minor) Supp page 43 "We design a ILP" -> "We design an ILP". Same on supp page 44."

We corrected those accordingly.

"(Minor) Is one of "the" and "our" redundant in "A detailed description of our the model selection procedure is in Supplementary Note B.4." ?"

We corrected it accordingly.

References

1. Makohon-Moore, A.P., Zhang, M., Reiter, J.G., Bozic, I., Allen, B., Kundu, D., Chatterjee, K., Wong, F., Jiao, Y., Kohutek, Z.A. and Hong, J. Limited heterogeneity of known driver gene mutations among the metastases of individual patients with pancreatic cancer. *Nature genetics*, 49(3): 358 (2017).
2. The Cancer Genome Atlas Research Network. Integrated genomic characterization of pancreatic ductal adenocarcinoma. *Cancer cell*, 32(2): 185-203 (2017).
3. Kim, C., Gao, R., Sei, E., Brandt, R., Hartman, J., Hatschek, T., Crosetto, N., Foukakis, T. and Navin, N.E. Chemoresistance evolution in triple-negative breast cancer delineated by single-cell sequencing. *Cell*, 173(4): 879-893 (2018).
4. Casasent, A.K., Schalck, A., Gao, R., Sei, E., Long, A., Pangburn, W., Casasent, T., Meric-Bernstam, F., Edgerton, M.E. and Navin, N.E. Multiclonal invasion in breast tumors identified by topographic single cell sequencing. *Cell*, 172(1-2): 205-217 (2018).
5. Zaccaria, S., El-Kebir, M., Klau, G.W. and Raphael, B.J. Phylogenetic copy-number factorization of multiple tumor samples. *Journal of Computational Biology*, 25(7): 689-708 (2018).

REVIEWERS' COMMENTS:

Reviewer #3 (Remarks to the Author):

I was satisfied with the previous version. I see that the authors have done an impressive job in responding carefully to all the points for a new set of comments.

Reviewer #4 (Remarks to the Author):

I thank the authors for considering my comments and modifying the manuscript.

Being quite familiar with the published work from the group of prof. Ben Raphael, I was very upset and disappointed with surprisingly large number of mathematical mistakes of all sorts present in the previous version of the manuscript. I hope that we have now found and corrected most of them and that this will make the whole publication better. It is also good that several overblown statements have now been relaxed or better clarified.

One point that remains is the following: I still do not completely understand the major difference between HATCHet and CNT-MD. Could the authors please provide more detailed clarification, including a clear examples where one should opt for using CNT-MD rather than HATCHet (and vice versa). I assume that there should be examples for both scenarios since in the Response to reviewers letter it is stated that HATCHet does not supersede CNT-MD (I will assume that CNT-ND used in the response letter is yet another typo). Why can't a user just run CNT-MD as it even provides an insight into tumor evolution?

Assume that I am interested in obtaining both, clonal composition of tumor and phylogenetic relationships of the clones. Are these two methods sufficient for getting some answer to this question? If yes, what should I use? Should I first run HATCHet and then CNT-MD, or can I just directly run CNT-MD? It is clear that I can not only run HATCHet as it does not perform phylogenetic inference.

I am bringing this up because I am concerned that some of potential users will unnecessarily cope with arduous task of figuring out which tools from this group are outdated and which ones should be tried out. In addition, lack of clear instructions about this can even reduce the number of users of HATCHet.

Can we also confirm that my understanding of the following is correct: (i) THetA and THetA2 are now completely obsolete and should not be used in practice at all in the future; and (ii) this will be indicated very clearly on the webpages of these tools; ?

Otherwise, I am satisfied with the corrections made and additional analysis performed. In my opinion, the method has a potential to be used in some studies of intra-tumor heterogeneity and I

recommend it for publication (assuming that my above questions/concerns are addressed satisfactorily).

Reviewer responses for manuscript NCOMMS-20-03751A

Simone Zaccaria¹ and Benjamin J. Raphael^{1,*}

¹Department of Computer Science, Princeton University, Princeton, NJ 08540

*Correspondence: braphael@princeton.edu

We thank the reviewers for their detailed and thoughtful comments. In response to their final comments, we provide below the answers (blue text) to each reviewers' comment (black text). All references to sections and figures refer to the revised version of the manuscript, including the revision of the main text and the revision of the Supplementary Note.

Reviewer 3

"I was satisfied with the previous version. I see that the authors have done an impressive job in responding carefully to all the points for a new set of comments."

We thank the reviewer for the positive evaluation of our manuscript.

Reviewer 4

"I thank the authors for considering my comments and modifying the manuscript. Being quite familiar with the published work from the group of prof. Ben Raphael, I was very upset and disappointed with surprisingly large number of mathematical mistakes of all sorts present in the previous version of the manuscript. I hope that we have now found and corrected most of them and that this will make the whole publication better. It is also good that several overblown statements have now been relaxed or better clarified."

We have carefully reviewed the text and equations and removed some extraneous claims while shortening the manuscript per editorial requirements.

"One point that remains is the following: I still do not completely understand the major difference between HATCHet and CNT-MD. Could the authors please provide more detailed clarification, including a clear examples where one should opt for using CNT-MD rather than HATCHet (and vice

versa). I assume that there should be examples for both scenarios since in the Response to reviewers letter it is stated that HATCHet does not supersede CNT-MD (I will assume that CNT-ND used in the response letter is yet another typo). Why can't a user just run CNT-MD as it even provides an insight into tumor evolution?

Assume that I am interested in obtaining both, clonal composition of tumor and phylogenetic relationships of the clones. Are these two methods sufficient for getting some answer to this question? If yes, what should I use? Should I first run HATCHet and then CNT-MD, or can I just directly run CNT-MD? It is clear that I can not only run HATCHet as it does not perform phylogenetic inference.

I am bringing this up because I am concerned that some of potential users will unnecessarily cope with arduous task of figuring out which tools from this group are outdated and which ones should be tried out. In addition, lack of clear instructions about this can even reduce the number of users of HATCHet.”

HATCHet infers integer and fractional copy numbers that can be used as input for reconstructing a phylogenetic tree using CNT-MD. Specifically, CNT-MD cannot be directly applied to DNA sequencing data but it requires an estimate of the fractional copy numbers, which is one of the main innovations of HATCHet. Therefore, if one is interested in inferring both the tumor clonal composition and tumor phylogeny, HATCHet and CNT-MD can be jointly used to solve these tasks, with HATCHet estimating the copy numbers and CNT-MD reconstructing their evolution. We added a related note in Section 4.4 of Method:

“We note that HATCHet does not directly reconstruct a tumor phylogenetic tree. However, the copy numbers inferred by HATCHet can be used as input to methods for phylogenetic reconstruction. For example, the integer copy numbers inferred by HATCHet can be input to MEDICC⁵¹ or CNT^{52,53}, and the fractional copy numbers can be input to CNT-MD^{23,24} or Canopy³⁷.”

Finally, we highlight that HATCHet infers allele-specific copy numbers, while CNT-MD only considers total copy numbers. Therefore, the allele-specific analysis of HATCHet is necessary to identify allele-specific CNAs, which include fundamental and frequent events in cancer like copy-neutral loss-of-heterozygosity (LOH) and whole-genome duplications (WGDs).

“Can we also confirm that my understanding of the following is correct: (i) THetA and THetA2 are now completely obsolete and should not be used in practice at all in the future; and (ii) this will be indicated very clearly on the webpages of these tools; ?”

We confirm the statements of the reviewer and we clearly added the following statement to the THetA's repository:

“UPDATE: If you aim to infer allele- and clone-specific copy-number aberrations (CNAs) from bulk tumor samples, we recommend that you use [HATCHet](<https://github.com/raphael-group/hatchet>), an new algorithm with several improvements over THetA.”

Otherwise, I am satisfied with the corrections made and additional analysis performed. In my opinion, the method has a potential to be used in some studies of intra-tumor heterogeneity and I recommend it for publication (assuming that my above questions/concerns are addressed satisfactorily).”

We thank the reviewer for the positive evaluation of our manuscript.